# Thermodynamic insight into stimuli-responsive behaviour of soft porous crystals

L. Vanduyfhuys[1], S.M.J. Rogge [1], J. Wieme [1], S. Vandenbrande[1], G. Maurin[2], M. Waroquier[1] & V. Van Speybroeck [1]

Knowledge of the thermodynamic potential in terms of the independent variables allows to characterize the macroscopic state of the system. However, in practice, it is difficult to access this potential experimentally due to irreversible transitions that occur between equilibrium states. A showcase example of sudden transitions between (meta)stable equilibrium states is observed for soft porous crystals possessing a network with long-range structural order, which can transform between various states upon external stimuli such as pressure, temperature and guest adsorption. Such phase transformations are typically characterized by large volume changes and may be followed experimentally by monitoring the volume change in terms of certain external triggers. Herein, we present a generalized thermodynamic approach to construct the underlying Helmholtz free energy as a function of the state variables that governs the observed behaviour based on microscopic simulations. This concept allows a unique identification of the conditions under which a material becomes flexible.

[1] Center for Molecular Modeling, Ghent University, Technologiepark 903, 9052 Zwijnaarde, Belgium. [2] Institut Charles Gerhardt Montpellier, Université Montpellier, Place E. Bataillon, 34095, Montpellier, Cedex 05, France. Correspondence and requests for materials should be addressed to L.V. (email: Louis.Vanduyfhuys@UGent.be) or to V.V.S. (email: Veronique.VanSpeybroeck@UGent.be)

At first, it might seem rather counterintuitive to connect flexibility and crystallinity. A crystalline phase is typically associated with a high degree of order and close packing of its components, which seemingly conflicts with softness or flexibility. However, new generations of materials have been discovered that, unlike most conventional inorganic crystals, possess a highly ordered network but are also able to structurally transform while retaining the same or similar topologies[1–3]. Kitagawa and colleagues[1] coined the term 'soft porous crystals' (SPCs) for such materials, which show a bistable or multistable behaviour with long-range structural order. Since then, an enormous endeavour has been undertaken to understand this experimentally observed phenomenon[4, 5]. Prominent examples of such SPCs are flexible metal-organic frameworks (MOFs), built up from inorganic nodes or chains connected with organic linkers, which have the ability to undergo drastic changes in the unit cell volume upon external stimuli (Fig. 1)[3]. Initially, it was thought that framework flexibility could only be induced by adsorbing or desorbing guest molecules; however, various materials have been discovered that show stimuli-responsive behaviour under the influence of external triggers such as mechanical pressure, temperature, electric fields and light[3, 6]. The term 'flexibility' is a rather generic term and may imply different forms of structural changes such as ligand flipping, pore gating, breathing, etc. In this work, it will be used to refer to materials with the ability to undergo phase transformations accompanied by a substantial change in unit cell volume and for which the crystallographic space group of the different phases may change (schematically shown in Fig. 1)[3]. Frameworks exhibiting this type of flexibility have often been categorized as breathing materials. For more elaborate discussions on the terminology we refer to the review of Schneemann et al.[3]. The prototype flexible frameworks are the so-called MIL-53 materials (MIL = Matériaux de l'Institut Lavoisier) exhibiting a wine-rack topology

and capable of undergoing volume changes up to 40% upon exposure to external stimuli[7]. The observed macroscopic flexibility has been found to be critically dependent on the detailed composition of the framework and the building units making up the framework (Fig. 1)[3, 4].

Experimentally, framework flexibility can be followed by monitoring the response of the material to the external stimulus (schematically shown in Fig. 2) with techniques such as mercury-intrusion porosimetry[8], high-pressure X-ray diffraction[9], differential scanning calorimetry[10] or spectroscopic techniques[11]. These approaches have been applied to a diverse set of soft porous materials and the observation of a sudden volume change at a given value of the external trigger uniquely identifies framework flexibility. Although the experimental profiles allow one to recognize the stimuli-responsive behaviour, they do not allow one to construct the underlying thermodynamic potential governing the observed behaviour. Indeed, during the breathing phenomenon, the material undergoes one or more irreversible structural transitions between (meta)stable equilibrium states, which are separated by a region of volume states, which are not equilibrated with the applied experimental conditions. These states are only visited during irreversible transitions and hence one cannot extract their equilibrium thermodynamic potential from experiment using basic equilibrium thermodynamics.

Yet, knowledge of the full thermodynamic potential in terms of the independent variables of the system is the key ingredient to understand the macroscopic conditions governing breathing. Within thermodynamics, a thermodynamic potential is defined as a function of certain variables characterizing the state of the system, having the property of being in a maximum or a minimum when the system is in equilibrium. Various thermodynamic potentials may be used such as the internal energy, the Helmholtz free energy and the Gibbs free energy. For systems characterized by the temperature, volume and number of particles as state

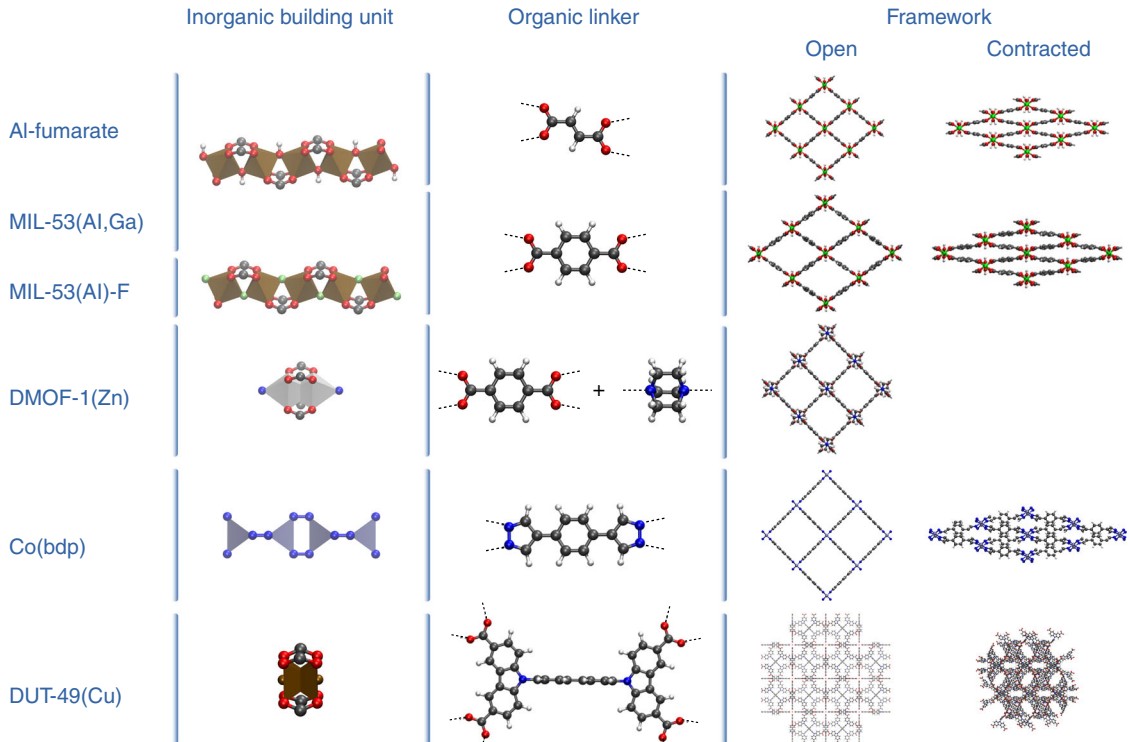

**Fig. 1** Illustration of some metal-organic frameworks which show potential stimuli-responsive behaviour. All shown materials are MOFs built from a 0D or 1D inorganic moieties connected by organic linkers

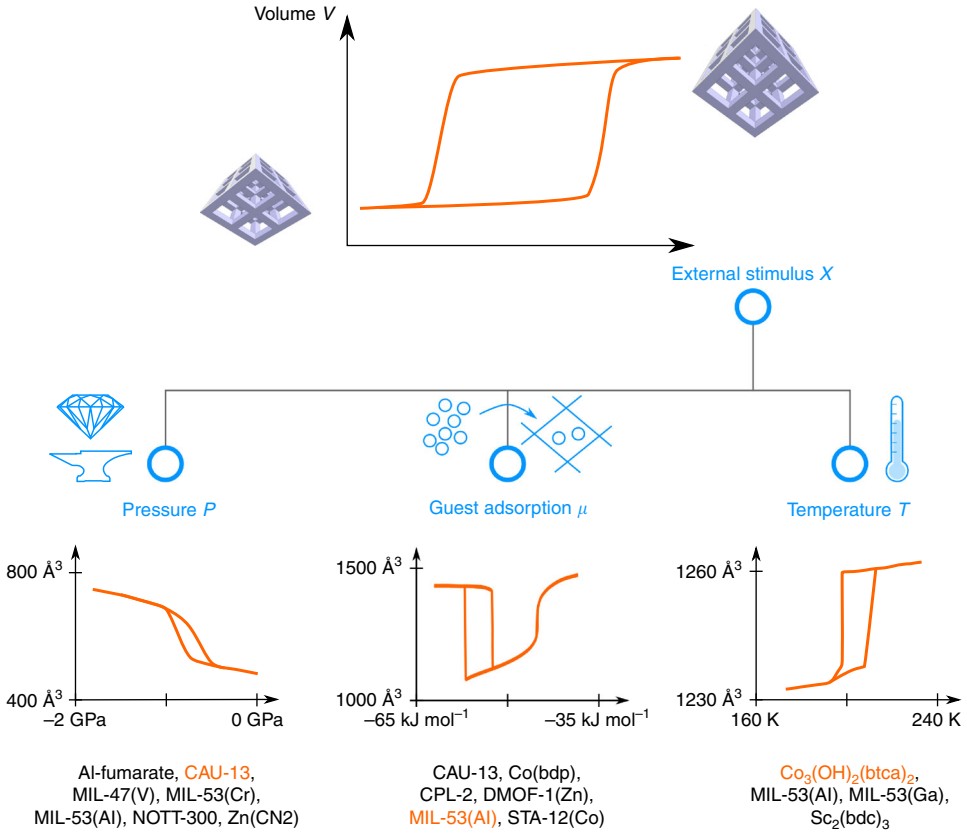

**Fig. 2** Illustration of volume profiles for three types of external stimuli. The stimuli are mechanical pressure, chemical potential and temperature. Examples of soft porous materials for which such profiles have been determined are indicated in the bottom of the figure (Al-Fumarate[17], CAU-13[48], MIL-47(V)[18], MIL-53(Cr)[49], NOTT-300[48], Zn(CN)$_2$[50], Co(bdp)[35], CPL-2[51], DMOF-1(Zn)[36], MIL-53(Al)[12, 19, 52], STA-12(Co)[53], Co$_3$(OH)$_2$(btca)$_2$[54], MIL-53(Ga)[26] and Sc$_2$(bdc)$_3$[55]). The shown profiles correspond, from left to right, to CAU-13, MIL-53(Al) and Co$_3$(OH)$_2$(btca)$_2$. For each material, the chemical formula and building blocks are given in Supplementary Table 1

variables, the Helmholtz free energy is introduced as follows:

$$dF = -SdT - PdV + \sum_i \mu_i dN_i \qquad (1)$$

However, one can transform one thermodynamic potential to another by means of a Legendre transformation[12].

Herein we present a microscopic approach based on classical molecular dynamics simulations to construct the Helmholtz free energy and to uniquely determine the macroscopic response of a material upon stimuli such as mechanical pressure, temperature and adsorbed guest molecules. The systematic thermodynamic approach allows for the identification of the macroscopic conditions under which a material is expected to undergo a structural transition. Some materials show rigidity in a certain window of state variables, while possessing intrinsic bistable or multistable behaviour characterized by a thermodynamic potential bearing one or more minima in terms of volume. Such materials may become flexible under another set of conditions. The approach is generic and may be applied directly to any SPC exhibiting long-range order. Materials for which the phase transition is accompanied by a loss of long-range order, such as amorphous MOFs[13], cannot be straightforwardly simulated as the present approach relies on periodic boundary conditions. Effects such as amorphization under elevated pressures may however be detected both experimentally and theoretically by a broadening of the peaks in the radial distribution function[14].

## Results

**Structural transitions induced by mechanical pressure**. From a thermodynamic point of view two cases need to be distinguished: (i) structural transitions induced by a mechanical pressure $P$ and (ii) structural transitions induced by another trigger $X \neq P$[15, 16]. The thermodynamic approach is introduced for the case where structural transitions are induced by application of a mechanical pressure under fixed conditions of number of particles and temperature. As the volume is the conjugate variable of the pressure, this is thermodynamically the simplest case. In general, the experimental response of a material under a positive mechanical pressure can yield three profiles, as illustrated in Fig. 3[17–21]. Negative pressures would correspond to pulling the crystal framework isotropically. Such conditions might become accessible experimentally by embedding the materials in a membrane; however we are not aware of the development of such a setup to test the material's behaviour at hand. In the first case (Type I in Fig. 3), the volume-versus-pressure profile shows no hysteresis and only a gradual decrease of the volume without structural transition is observed upon increasing pressure. The thermodynamic cycle is reversible upon exerting an external pressure and releasing it afterwards. The material behaves as an ideal spring, where both the material and the environment return to their initial state after the closed pressure cycle. Most conventional solids belong to this category in the elastic regime. In the second case (Type II in Fig. 3), the material suddenly switches towards a contracted pore phase but never returns to the original open pore phase even when releasing the pressure. The third case

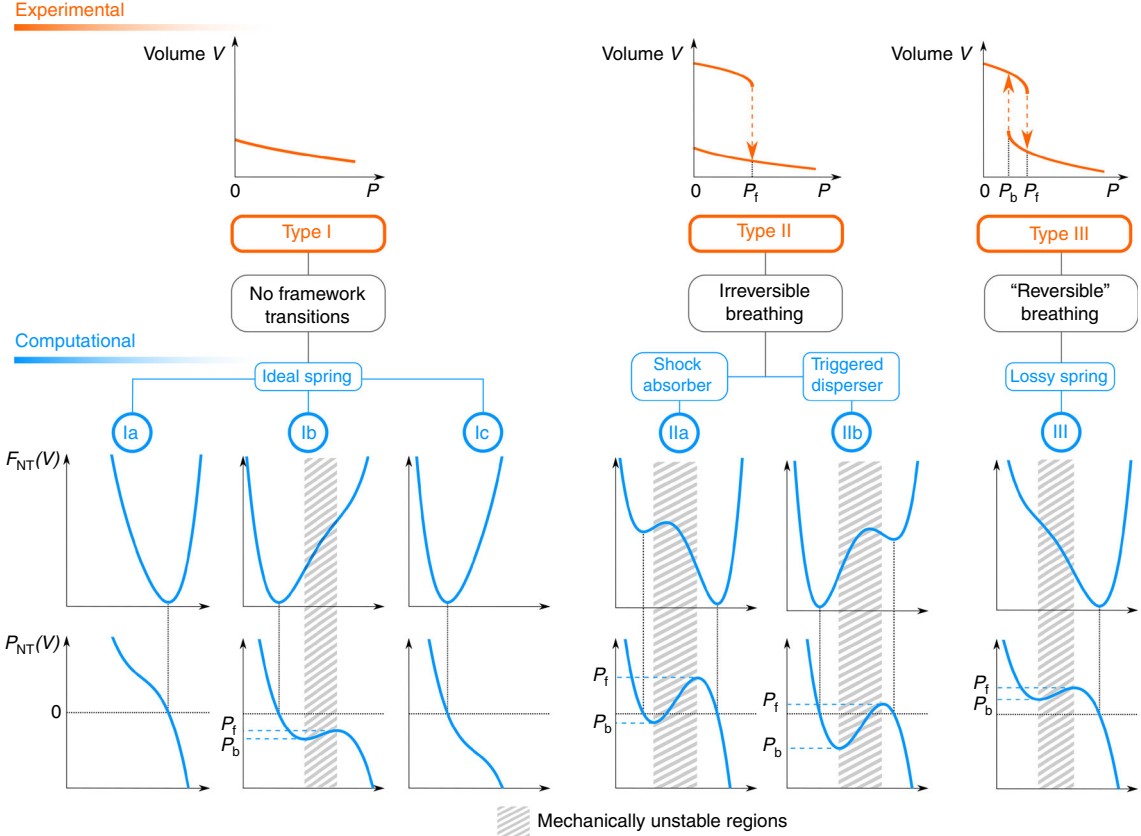

**Fig. 3** Comparison of experimental response with corresponding hypothetical free energy. Top panel: illlustration of the possible experimental responses of a material upon exerting a positive mechanical pressure[17–21]. Bottom panel: hypothetical Helmholtz free energy $F_{NT}(V)$ and mechanical equation of state $P_{NT}(V)$, with $N$ the number of particles and $T$ the temperature, in terms of the volume. Shaded areas correspond to mechanically unstable regions

(Type III in Fig. 3) corresponds to a back-and-forth transition from one phase to another when increasing or releasing the pressure. In this case hysteresis is observed in the volume-versus-pressure profile. Although the sketched experimental $V(P)$ profiles unequivocally define the material's response to external pressure, they do not allow one to uniquely assess the underlying thermodynamic potential.

The same volume-versus-pressure profile may result from fundamentally different thermodynamic potentials, as it will be illustrated hereafter. For systems where only the PV work is relevant, i.e., isotropic or anisotropic systems exposed to a hydrostatic pressure, the state of the system is uniquely defined by the Helmholtz free energy $F_{NT}$ as a function of the number of particles $N$, volume $V$ and the temperature $T$. As the hydrostatic pressure is the negative volume derivative of the Helmholtz free energy, one can obtain the thermodynamic potential $F_{NT}$ relative to some reference point using thermodynamic integration:

$$F_{NT}(V) - F_{NT}(V_{ref}) = \int_{V_{ref}}^{V} \frac{\partial F_{NT}(V')}{\partial V'} dV' = -\int_{V_{ref}}^{V} \langle P_{NT}(V') \rangle dV'$$

(2)

where $P$ is the instantaneous hydrostatic pressure[22]. The latter quantity may be obtained from atomistic molecular dynamics simulations where the volume is constrained but the cell shape is allowed to fluctuate as explained in the Methods section. This procedure yields the mechanical equation of state $P_{NT}(V)$, for which six distinct profiles can be distinguished, as displayed in the bottom pane of Fig. 3. Via thermodynamic integration each subtype of $P_{NT}(V)$ results in a specific shape of the free energy

$F_{NT}(V)$. In all plots the shaded areas correspond to mechanically unstable regions, where $\frac{\partial P}{\partial V} > 0$ or, equivalently, with regions having a negative bulk modulus, which are not accessible experimentally. A given experimental response may originate from fundamentally different free energy curves, as schematically shown in Fig. 3. To properly understand the possible applications of the material under a given set of experimental conditions, it is crucial to distinguish between the various free energy profiles. For example, materials bearing a Type II behaviour may either be used as a shock absorber or a triggered disperser. We will now illustrate the approach to a series of MOFs, to predict their flexibility characteristics. For all materials, we constructed the thermodynamic potential from molecular dynamics simulations using in-house developed force fields (Methods section). The mechanical equations of state and the free energy profiles are shown in Fig. 4a.

UiO-66 and MIL-53(Ga) materials both behave as an ideal spring and show a volume-versus-pressure response of the Type I. UiO-66 is the prototype example of a very stable MOF under various conditions and is characterized by a Helmholtz free energy profile belonging to the Type Ia, as it has one stable minimum and no inflection points (see Supplementary Fig. 1)[14,23–25]. Experimentally, the material shows a loss of crystallinity only at a hydrostatic pressure of about 1.4 GPa and the material is characterized by a high bulk modulus of about 17 GPa[20]. MIL-53 (Ga) on the contrary belongs to the well-known MIL-53 series featuring a typical wine-rack framework which is a characteristic for many flexible materials[21,26]. Structural transitions between an open pore and a contracted pore phase were observed under influence of temperature or guest adsorption, but the pressure response is to date unknown. A full tensorial analysis of the elastic

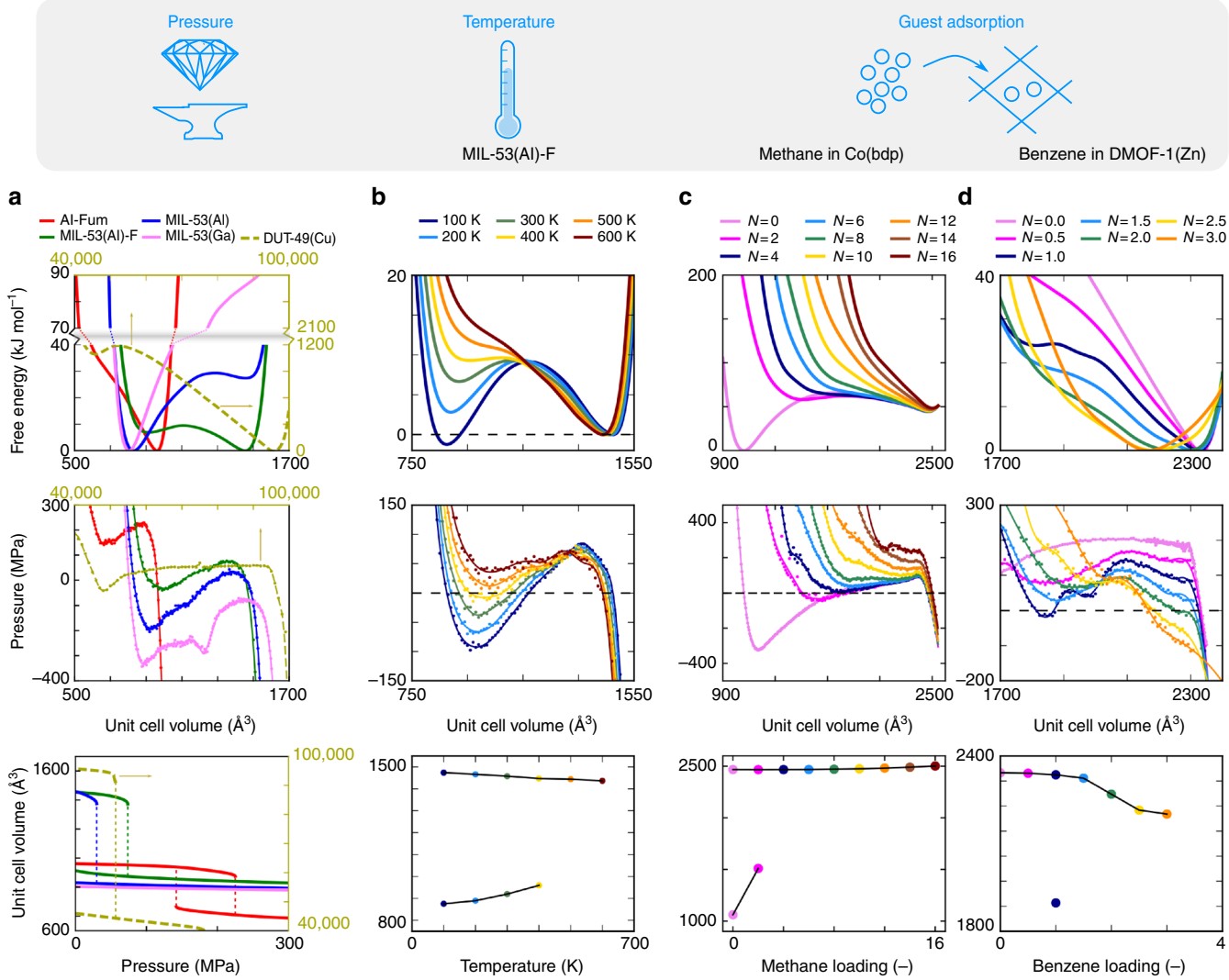

**Fig. 4** Illustration of the mechanical equations of state for a variety of materials. Upper row: free energy profile; middle row: mechanical equations of state; bottom row: volume versus external trigger. **a** Free energies, mechanical equations of state $P_{NT}(V)$ and V(P) profiles for Al-Fumarate, MIL-53(Ga), MIL-53 (Al)-F, MIL-53 and DUT-49(Cu) (DUT-49(Cu) axis are shown in olive green). **b** Free energies, mechanical equations of state $P_N(V;T)$ and V(T) profiles for MIL-53(Al)-F. **c,d** Free energies, mechanical equations of state $P_T(V;N)$ and V(N) profiles for methane@Co(bdp) and benzene@DMOF-1(Zn)

constants was performed by Ortiz et al.[27]. They found a strong anisotropic behaviour of the Young modulus in various directions, which is an indicator for potential flexible behaviour. While such analysis of the anisotropy points towards potential bistable behaviour of a material, it does not reveal the conditions under which the material breathes[28]. The thermodynamic potential—belonging to Type Ib—shows various inflection points, which is necessary for flexible behaviour. However, as the Helmholtz free energy reveals no stable minimum for the open pore phase, the material is predicted to remain in its contracted pore phase upon exerting positive pressure and releasing it back.

The functionality of the MIL-53 type of materials may be modulated by changing the linkers and/or metals. The MIL-53 (Al) and the MIL-53(Al)-F are both isoreticular to the MIL-53 (Ga) but belong to the Type II class upon response to a positive mechanical pressure[19,29]. MIL-53(Al) and MIL-53(Al)-F are both constructed with aluminum in the inorganic chains, but MIL-53 (Al) contains $OH^−$ compensating anions, whereas MIL-53(Al)-F has $F^−$ charge-compensating anions in the inorganic chains (Fig. 1)[29]. These two materials are predicted to show an irreversible pressure-induced breathing. When increasing the pressure, the material, which is originally in its open pore phase,

will irreversibly transform to a contracted pore phase at a pressure $P_f$. When decreasing the pressure back to zero, the material relaxes reversibly to its contracted pore phase, but never returns to the open pore phase. According to the applied force fields, the Helmholtz free energy profiles are fundamentally different in both cases. For MIL-53(Al), the open pore phase is a metastable state (Type IIb) and the material behaves as a triggered disperser. During a compression/decompression cycle, the material releases part of its energy as heat to the environment, effectively increasing the entropy of the environment and as such also the universe. The pressure-induced open pore phase to contracted pore phase transition was experimentally observed at pressures of 13–18 MPa[19,22]. The material also shows a large anisotropy of the Young modulus, which is a necessary criterion for breathing[27]. Similar pressure-induced flexibility is predicted from our model for the Co(bdp) material, not belonging to the MIL-53 series[30]. Current experimental procedures only allowed to synthesize a fully desolvated structure in the contracted pore phase, whereas our model predicts the existence of a large and contracted empty pore phase, which is only slightly less stable (45 kJ $mol^{-1}$ at 300 K). This material might be interesting for further testing in a compression/decompression cycle. In contrast, MIL-

53(Al)-F shows a free energy profile of Type IIa and behaves as a shock absorber and is potentially interesting for energy storage applications. Starting from the open pore phase and applying an external pressure, the material will also irreversibly contract towards the contracted pore phase, but in this particular case the material will absorb work resulting in an increase of the free energy ($\Delta F_{MOF} > 0$). According to the first and second law of thermodynamics, the total work performed on the MOF equals $W_{MOF} = \Delta F_{MOF} + T\Delta S_{universe} > \Delta F_{MOF}$ and it is larger than the change in free energy of the MOF, since the entropy of the universe will rise during an irreversible process. The $\Delta F_{MOF}$ is a lower bound for the work required to induce the open pore phase to the contracted pore phase transition. To maximize the stored mechanical energy, the MOF should preferentially exhibit a large volume difference between the two (meta)stable states and a high transition pressure. DUT-49(Cu) is another example of a potential shock absorber (Fig. 4a)[31]. According to our simulations, DUT-49(Cu) is predicted to have the ability to store more mechanical energy as MIL-53(Al)-F (57.8 J g$^{-1}$ vs 7.9 J g$^{-1}$). This might seem contradictory, as the transition pressure of DUT-49 (Cu) is lower; however, as explained in Supplementary Note 3, the larger volume variation (per gram material) is the main cause for the higher amount of absorbed mechanical energy.

As an example of a Type III material, we discuss the Aluminum fumarate MOF (Al-fumarate), which shows highly flexible behaviour. The material is isoreticular to the MIL-53(Al) but contains aliphatic fumarate linkers instead of the aromatic benzenedicarboxylate linkers (Fig. 1)[17,32]. Our thermodynamic model predicts that, during a compression/decompression cycle, the material switches to its contracted pore phase once the pressure $P_f$ is reached and returns to its open pore phase when releasing the pressures to values lower than $P_b$. This is fully consistent with the recent experimental observation which revealed that the reversible contraction of this material to the contracted pore phase occurs at a high transition pressure of about 120 MPa and returns back to the open pore phase upon releasing the pressure below 20 MPa[17]. The amount of mechanical energy that could be stored in one compression–decompression cycle was unprecedently high at that time with values estimated to be of the order of 40–60 J g$^{-1}$[17]. Within the class of porous solids, this material has large potential to be used as nano-dampener or lossy spring.

**Structural transitions induced by a trigger $X \neq P$.** The strength of our thermodynamic approach lies in the fact that it may also predict the response of the material upon other stimuli such as the temperature or the chemical potential, which are not conjugate variables of the volume. To investigate transitions induced by such an external trigger $X$, a series of pressure versus volume profiles $P_Y(V;X)$ for various values of the external trigger $X$ need

to be constructed while the other state variables $Y$ are kept constant (Fig. 5a). The principle is conceptually illustrated in Fig. 5. The volume response of the material upon the external stimulus $X$ is obtained by determining for each $X$ value of the volume $V$ for which $P(V;X) = P_0$ and $\frac{\partial P}{\partial V} < 0$ from its mechanical equation of state. This is done by intersecting the mechanical equations of state at a given pressure $P = P_0$ (usually the atmospheric pressure), from which the $V(X)$ profiles are obtained (Fig. 5c). This $V(X)$ profile contains all information to deduce whether a material is expected to undergo a structural transformation under the influence of the external trigger $X$.

There are ample cases available for which breathing transitions were observed experimentally under influence of temperature and/or guest adsorption. For a selected set of materials, we constructed the mechanical equations of state in terms of various temperatures and/or guest loadings. The results are shown in panels b, c and d of Fig. 4. The thermo-responsive behaviour is illustrated for MIL-53(Al)-F in Fig. 4b. Experimentally, no temperature-induced transition to the contracted pore phase was observed, only a phase transition from an orthorhombic open pore phase to a monoclinic open pore phase at 175 K[29]. A series of molecular dynamics simulations in the temperature window 100–600 K were performed to generate the $V(T)$ profile. Our thermodynamic approach reveals that the material exhibits two (meta)stable structures at temperatures below 400 K, whereas only one stable structure is found above 500 K. A transition to the contracted pore phase is predicted to be only possible if the open pore phase minimum in the free energy profile would disappear, according to the assumption of collective behaviour[33]. Our model predicts that when starting from the contracted pore phase at low temperatures, a structural transition to an open pore phase would take place between 400 and 500 K. This experiment was not performed so far. Nanthamathee et al.[29] determined the volumetric thermal expansion coefficient $\alpha_V = \frac{1}{V}\left(\frac{\partial V}{\partial T}\right)_P$ in the temperature range 175–500 K, which was found to be negative for MIL-53(Al)-F in contrast to its -OH analogue. The $V(T)$ profile allows to directly deduce the thermal expansion coefficient in both the contracted and open pore phases. A negative coefficient is indeed found in the open pore phase, whereas the lower branch corresponding to the contracted pore phase is characterized by a positive thermal expansion coefficient. In fact, the volumetric thermal expansion coefficient $\alpha_V$ is a local descriptor of framework distortions but gives by itself no conclusive outcome on the potential phase transformations of the material in a given temperature window.

Guest-induced phase transformations are illustrated for xenon adsorption in MIL-53(Al) (Supplementary Fig. 2), methane adsorption in Co(bdp) (Fig. 4c) and benzene adsorption in DMOF-1(Zn) (Fig. 4d). We investigated the response of all these materials upon guest loading by constructing a series of $P_T(V; N)$

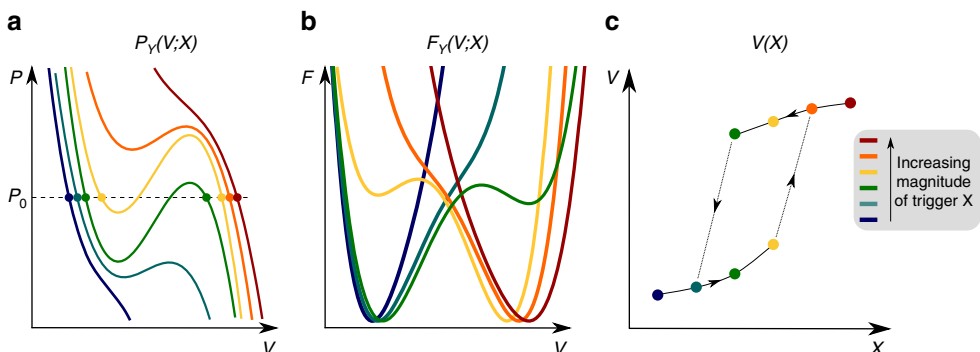

**Fig. 5** Illustration of the structural transitions induced by a general state variable $X$. **a** Mechanical equation of state **b** free energy and (**c**) the response $V(X)$

mechanical equations of state for a fixed temperature but with a varying number of particles (see Methods section). Experimentally, one controls the chemical potential; however, computationally, it is easier to perform molecular dynamics simulations by controlling the number of particles. The obtained $P_T(V; N)$ profile can easily be transformed to a $P_T(V; \mu)$ equation of state by means of a Legendre transformation of the corresponding Helmholtz free energy to the grand canonical potential[12]. By localizing the stable points at the intersection of the pressure profiles with a fixed mechanical pressure of 1 bar, the $V(N)$ response curve upon guest loading are constructed (Fig. 4c for methane@Co(bdp) and Supplementary Fig. 2 for xenon@MIL-53(Al)). For xenon adsorption in MIL-53(Al), our model predicts that the material undergoes a phase transition to an intermediate phase with a volume in between the contracted pore and open pore phases, when initially starting from the open pore phase, for a loading of one xenon particle per unit cell. Upon higher loading the volume gradually increases. These results agree well with the experiments of Boutin et al.[34], which evidenced breathing transitions in the measured xenon adsorption isotherms for MIL-53(Al) in the temperature range 195–323 K and earlier simulations based on a mean-field model[12, 34]. For Co(bdp) we predict a transition from a contracted pore phase to an open pore phase for a methane loading varying between two and four molecules, which agrees well with the experiments of Mason et al.[35]. The proposed model is versatile in the sense that not only phase transformations may be predicted but also continuous shrinking of the material as was observed for benzene adsorption in DMOF-1(Zn), which is a pillared layer MOF (Fig. 1)[36, 37]. In this case a continuous shrinking of the open pore phase is observed upon loading with benzene molecules (Fig. 4d). The transition is observed from a loading from about 1.5 to 2 benzene molecules per unit cell. Our observations are in correspondence with the results of Grosch et al.[36] who performed a series of *NPT* simulations, but did not extract the full thermodynamic potential. The free energy determined here, reveals that a local minimum exists, which is, however, not sufficiently stabilized by the guest molecules to induce a phase transformation. Our thermodynamic model yields all necessary components to also explain unexpected effects such as negative gas adsorption, which was observed for DUT-49(Cu) upon exposure of methane, as explained in Supplementary Note 5[38]. The approach proposed here, might be helpful to design materials with specific guest induced breathing in close collaboration with experimentalists[39].

## Discussion

In conclusion, we presented a generalized thermodynamic approach to construct the Helmholtz free energy as a function of the state variables for SPCs based on microscopic simulations. The method relies on the construction of the mechanical equation of state $P_Y(V;X)$ from molecular dynamics simulations for various natures and magnitudes of the external trigger $X$. In case the external stimulus is the mechanical pressure, the response of the material at a given temperature may be easily derived from one mechanical equation of state. For more general stimuli such as the temperature or the chemical potential of adsorbing guest molecules, which are not the conjugate variable of the volume, a series of mechanical equations of state at various values of $X$ need to be determined. As such, one obtains the full thermodynamic potential in terms of the governing state variables and a complete characterization of the framework flexibility. The strength of the proposed thermodynamic model relies in the fact that it can predict a flexibility window for a given material in a generic way. We have shown that some materials, which show intrinsic bistable behaviour, may not adopt a flexible behaviour macroscopically.

This was the case for the MIL-53(Ga) for which local descriptors such as the anisotropy of Young modulus predict an intrinsic bistability, but which does not breathe under influence of pressure. Such local descriptors do not give a conclusive statement on the flexibility of the material under certain conditions. From an application point of view, knowledge of the thermodynamic potential is crucial, as it may allow the in silico anticipation of the selection of soft porous materials with adequate structural behaviour upon external stimuli, e.g., systems as shock absorbers or triggered dispersers when applying mechanical pressure.

## Methods

**Force field derivation**. Molecular simulations for each MOF were performed using a force field specifically derived for each MOF. For DMOF-1(Zn), including the benzene guest molecules, the force field derived by Grosch et al.[36] was used. For MIL-53(Al), MIL-53(Al)-F, MIL-53(Ga), and Al-fumarate and Co(bdp), a force field was derived from periodic ab initio input using QuickFF, an in-house developed program to easily derive force fields for MOFs[40]. The required ab initio input was obtained by performing periodic Density Functional Theory calculations with the Vienna Ab Initio Simulation Package[41]. These force fields consist of a covalent contribution defined by QuickFF, an electrostatic contribution described by Coulomb interactions between atomic charges estimated by means of the Minimal Basis Iterative Stockholder partitioning scheme and a van der Waals contribution taken from the MM3 force field[42, 43]. More details on the construction of the force fields can be found in Supplementary Note 6. The thermodynamic model is in principle generic but relies on reliable force fields. As a typical illustration, we compared the results obtained for DMOF-1(Zn) with various force fields (Supplementary Note 7). For some materials such as one- and two-dimensional coordination polymers, force field development might be very challenging due to critical dependence on the non-bonding interactions between the layers or the chains and more dedicated research might be required to apply the methodology on these systems[44, 45].

**Construction of free energy profiles**. To construct the equation of state $P_{NT}(V)$, several molecular dynamics simulations were run in the $(N, V, \sigma_a = 0, T)$ ensemble for various values of the unit cell volume, whereas the cell shape may fluctuate. Herein, $\sigma_a$ is defined as the anisotropic contribution to the stress tensor. To mimic the experimental conditions of isotropic stress, we chose $\sigma_a = 0$, which is controlled by means of a Martyna–Tobias–Tuckerman–Klein barostat[46]. The number of particles is kept fixed and the temperature is controlled using the Nosé–Hoover thermostat[47]. From the output of such molecular dynamics simulations, the time average of the instantaneous hydrostatic pressure $P_i$ was computed, which is defined at each time step as $P_i = Tr(\sigma_i)/3$ with $\sigma_i$ the instantaneous, internal stress tensor resulting in $P(V)$ and $F(V)$ profiles. More information about the procedure can be found in the work of Rogge et al.[22]. The details of the computational setup of the molecular dynamics simulations can be found in Supplementary Note 8. For a selected set of materials, the separate contribution of internal energy and entropy was deduced from the simulations to obtain more insight into the effects contributing to the stabilization of one or the other phase (Supplementary Note 9). Alternatively, free energy profiles may be obtained in the quasi-harmonic approximation. This was done for some materials under study in this paper, more information can be found in Supplementary Note 10. Finally, the convergence of the free energy profiles in terms of the simulation time is investigated in Supplementary Note 11.

**Data availability**. The authors declare that all data supporting the findings of this study are available within the paper and the Supplementary Files, or available from the authors upon request (see Supplementary Note 12).

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

## Acknowledgements

We acknowledge the Fund for Scientific Research—Flanders (FWO), the Research Board of Ghent University (BOF), BELSPO, in the frame of IAP/7/05 for financial support. V.V. S. acknowledges funding from the European Union's Horizon 2020 research and innovation programme (consolidator ERC grant agreement number 647755–DYNPOR (2015–2020)). The computational resources and services used in this work were provided by VSC (Flemish Supercomputer Center), funded by the Hercules foundation and the Flemish Government—department EWI. We also thank J.D. Evans and F.-X. Coudert for providing us the data of the $F_T(V;N)$ profiles from their simulations of methane in DUT-49(Cu).

## Author contributions

L.V., S.M.J.R., J.W., S.V., G.M. M.W., and V.V.S initiated the discussion and designed the paper. V.V.S. and L.V. wrote the manuscript with contributions of all authors. L.V., S.M. J.R., J.W., S.V., G.M., M.W. and V.V.S were involved in the discussion of the results and commented on the manuscript. J.W. performed the ab initio calculations. L.V. and J.W. constructed the force fields for all materials under study. L.V., S.M.J.R and S.V. carried out the molecular dynamics simulations.

## Additional information

**Competing interests:** The authors declare no competing financial interests.

