## [Peer Review File · Nature Communications]

Reviewer #1 (Remarks to the Author):

I was asked to review the manuscript by Vanduyfhuys and co-workers who developed a thermodynamic model that allows the computational study of the underlying thermodynamics in flexible MOFs. The target of the work, fundamental understanding of flexible MOFs, is highly topical and presents a fascinating research field of great current multidisciplinary (!) interest. Therefore, I regard the general topic of this field as highly suitable for Nat. Commun.

General considerations:

The target of the work itself, development of a model describing the underlying thermodynamics of the structural flexibility in MOFs is interesting. The study is well-organised, nicely written and the reader carefully guided through the development of the thermodynamic model of which the necessary calculations have been performed by force field based molecular simulations. Importantly, the model not only allows for quantifying the thermodynamics of an observed phase transition in a material, but further – and this the strength of the developed model– the identification of flexibility in new frameworks computationally.

My current concern directly follows up on the strength of the model, the prediction of flexibility upon a certain stimulus, which, however, has not been verified convincingly by experiment (yet). I agree with the authors that the developed model has a large potential, however, this potential need to be quantified for one or two typical and important flexible MOFs. In the current state, I feel that the study is better suited in a more specialised journal and therefore I am recommending rejection of the manuscript, however, with highly motivating the authors of resubmission. For the resubmission, the predictive strength of the model needs to be highlighted and verified, e.g. by application of the model to one or two exciting systems of recent literature, e.g. the M(bdp) series, or DUT-49.

Major points:

As mentioned above, the development of the model is performed in a nice way. The verification of the model to experimentally data, which has been done for a few members of the prototypical MIL series is similarly sound. The natural last step, which would justify publication is the application of the model to new flexible MOFs of which experimental investigation is not fully completed. Although this has been done in the manuscript for MIL-53(Al)-F regarding the high temperature phase transition, this is the only example given and not a particularly topical one. Exciting examples would be the application of the model to the M(bdp) series (pressure), as well as pressure to the DUT-49 (pressure). In this context, it will be also exciting to see if the negative gas adsorption behaviour of DUT-49 falls within one of the given categories.

Minor points:

Is it possible to investigate the role of entropy more explicitly? Particularly, the temperature driven flexibility is determined by the balance of dispersion interactions and lattice entropic gain, hence a study of entropy would be very interesting at least for $X = T$.

Figure1: for a non-expert in the field, the inorganic building unit shown in Figure 1 is somewhat confusing. I suggest updating the figures, in particularly the metal nodes and include the coordination environment of the metals.

Page 2, line 3 “techniques such as mercury-intrusion porosimetry, high-pressure X-ray diffraction, differential scanning calorimetry or inelastic neutron scattering”. Either add Terahertz spectroscopy at the end, or merge it together with inelastic neutron scattering to spectroscopic techniques. To my knowledge, yet there are no temperature/pressure dependent neutron scattering studies on flexible MOFs.

Typo at page 6, very top “remain remain”.

Reviewer #2 (Remarks to the Author):

The authors demonstrated the prediction and calculation of how "soft" behaviors of metal organic framework (MOF) crystals are. As a background, it is well known that a certain amount of MOF crystal show dynamic/flexible structural transformation upon external stimuli or guest accommodation. Many works are reported for each compound, and it is desired to generalize the flexible behavior for MOFs by the theoretical approach by use of force fields and other techniques with using of crystal structure. Because the approach could elucidate more useful insights of the flexible nature for other MOFs for future-application as the author mentioned in Figure 3 and the section of conclusion.

I read the manuscript carefully, and found the proposed results and discussion is of significant to understand the unique nature of soft MOF and realized that the approach is powerful to illustrate the different types of flexible behaviors in various MOFs. I found several new information which are not evaluated by the experiments, and the results herein support the experiments and prediction of the flexibility-based functionality of MOFs. The paper mostly focuses on the three-dimensional structured MOFs but there are many flexible MOFs with two-dimensional or even one-dimensional coordination polymers (e.g. Nano Lett 2006, 6, 2581-2584.). I am not sure how the present approach could be available for the other flexible MOFs/CPs in terms of creation of force field, and the authors should add explanation about this point. I also wondered how the behavior of crystal-to-amorphous (reversible) transformation which is recently highlighted will be explained based on the proposed approach (e.g. Acc. Chem. Res. 2014, 47, 1555-1562. or APL Materials 2014, 2, 124401.). The authors only studied on the crystalline nature but the expansion of the discussion also for glass or amorphous MOFs would become more strong to generalize the work in here. At least the authors should mention/discuss the potential or perspective to access this issue in this manuscript.

This work has enough generality and novelty to explain/predict the flexible nature of various MOFs and it would contribute to explore the new functions and application in the future which would be worth to be published in this journal, but I address some points to be revised to consider the final decision as following.

1. The equation in page 2 (last part) should have an explanation for the term of $\Sigma(\text{uidNi})$.
2. Figure 2 has many MOF names but many of them are not dealt with in this paper. NOTT-300, $\text{Zn}(\text{CN})_2$, STA-12(Co) etc. The information confuses the readers and I recommend them to be moved to the supporting information or delete. Also there are many abbreviation of MOFs and it is better to describe the precise formula of each MOF in the main manuscript, otherwise it is not friendly to the non-MOF scientists. Figure 1 is not sufficient to let people understand the metal-ligand connectivity and assembled open structure, DMOF-1 has two ligands but the current Figure 1 does not have such information.
3. In principle they employed simple MOFs which means the structures have high symmetry, simple composition, not large cell. On the other hand, there are many important MOFs having low symmetry, large unit cell, and often various organic ligands are involved in one open structures (e.g. Angew. Chem. Int. Ed. 2010, 49, 4820-4824.). How about the (potential) solution to understand the flexible nature by the proposed method? In recent, mixed ligand approach is specifically important to tune the flexibility for application. Please provide some additional data or at least discussion.
4. References are required for the parts "...contracted phase occurs at high transition pressure of

about 120 MPa and returns back to the open phase upon releasing the pressure below 20 MPa." (page 6) or "...which was found to be negative for MIL-53(Al)-F in contrast to its OH analogue." (page 8).

If the authors response these points and provide appropriate revision or additional results, I consider the manuscript is acceptable for this journal.

Reviewer #3 (Remarks to the Author):

The manuscript reports on a systematic analysis of the thermodynamic behavior of flexible porous materials, in particular metal-organic frameworks, obtained through theoretical calculations based on molecular dynamics simulations.

The manuscript is well written and well organized.

The work that has been carried out leads to significant results of general and wide interest to the scientific community. I believe the paper is worthy to be published on Nature Communications but some revision is needed.

I have a few concerns about the theoretical calculations.

1) Authors used a force-field (FF) that has been developed specifically for treating flexible MOFs. As demonstrated by published articles, the FF has been carefully tested, but I wonder how results are affected by the functional form of the FF. Could they try to repeat some of the calculations with other "standard" FFs and see how they work with respect to the QuickFF.

2) Prediction of thermodynamic properties through molecular dynamics requires rather long simulations to be sure to have a reliable sampling of the PES. According to the supplementary information, authors run simulations for a given time. How can they be sure that the simulation time was long enough? Could they repeat the simulations for different time scales at least for one system?

3) Authors adopts a molecular dynamics (MD) approach to predict the thermodynamic behavior of the flexible systems, but they do not mention that the same information could be obtained through lattice dynamics (LD) in a quasi-harmonic approximation, for instance. To confirm their findings and their wider reliability, authors should also try to use LD. In that respect, LD is more suitable to ab initio quantum mechanical methods and not only FF. So, this could be an even stronger evidence of the results obtained with classical simulations.

4) In the manuscript, the first time the authors state that the work is based on molecular dynamics simulations is at page 5. Before that, they generically use "microscopic approach". That's true, but in my opinion, the level of theory adopted in the "microscopic approach" should be clearly specified at the beginning of the paper.

Reply to Reviewer #1

We thank the reviewer for his/her constructive comments. We were very pleased with the assessment of the reviewer about the topic of the work which was judged to be highly suitable for Nature Communications. The reviewer encouraged us to show even more the predictive power of the proposed thermodynamic model, by studying some typical and important MOFs for which no experimental data are available. We were highly triggered by this comment and decided to conduct an *in-depth* study on some of the most challenging highly flexible materials (Co(bdp) and DUT-49(Cu) – as outlined in detail further in the response letter) for which no complete experimental insight on their stimuli responsive behavior is available. We decided to focus on the two proposed MOFs to make the exploitation of our thermodynamic model even more decisive. Indeed, we used the thermodynamic model to predict the pressure-induced behavior of Co(bdp) and to rationalize one of the most spectacular and unexpected guest-assisted breathing behaviors of MOFs, *i.e.* the negative gas adsorption observed in DUT-49(Cu). Apart from this major point of the reviewer, we also improved the manuscript by taking into account all other raised comments. Overall, we are convinced that with the input of the reviewer, we were able to bring the manuscript to a substantially higher level. We appreciate his/her input and the constructive final assessment very much. We hope that the study is now suitable to be published in Nature Communications. Below the comments of the reviewer are copied (in italics) with an answer from our side.

I was asked to review the manuscript by Vanduyfhuys and co-workers who developed a thermodynamic model that allows the computational study of the underlying thermodynamics in flexible MOFs. The target of the work, fundamental understanding of flexible MOFs, is highly topical and presents a fascinating research field of great current multidisciplinary (!) interest. Therefore, I regard the general topic of this field as highly suitable for Nat. Commun.

General considerations:

The target of the work itself, development of a model describing the underlying thermodynamics of the structural flexibility in MOFs is interesting. The study is well-organised, nicely written and the reader carefully guided through the development of the thermodynamic model of which the necessary calculations have been performed by force field based molecular simulations. Importantly, the model not only allows for quantifying the thermodynamics of an observed phase transition in a material, but further – and this the strength of the developed model – the identification of flexibility in new frameworks computationally.

My current concern directly follows up on the strength of the model, the prediction of flexibility upon a certain stimulus, which, however, has not been verified convincingly by experiment (yet). I agree with the authors that the developed model has a large potential, however, this potential need to be quantified for one or two typical and important flexible MOFs. In the current state, I feel that the study is better suited in a more specialised journal and therefore I am recommending rejection of the manuscript, however, with highly motivating the authors of resubmission. For the resubmission, the predictive

strength of the model needs to be highlighted and verified, e.g. by application of the model to one or two exciting systems of recent literature, e.g. the *M(bdp)* series, or DUT-49(Cu).

Major points:

*As mentioned above, the development of the model is performed in a nice way. The verification of the model to experimental data, which has been done for a few members of the prototypical MIL series is similarly sound. The natural last step, which would justify publication is the application of the model to new flexible MOFs of which experimental investigation is not fully completed. Although this has been done in the manuscript for MIL-53(Al)-F regarding the high temperature phase transition, this is the only example given and not a particularly topical one. Exciting examples would be the application of the model to the *M(bdp)* series (pressure), as well as pressure to the DUT-49(Cu) (pressure). In this context, it will be also exciting to see if the negative gas adsorption behaviour of DUT-49(Cu) falls within one of the given categories.*

Response on the predictive power of the thermodynamic model and its application to some more typical and important flexible MOFs:

We studied both the stimuli responsive behavior of the Co(bdp) in absence and presence of methane molecules. For the methane-induced responsive behavior, experimental data are available, however, the pressure-induced flexibility of the empty frameworks has never been explored so far. Our thermodynamic model predicted that the Co(bdp) behaves as a triggered disperser when exposed to mechanical pressure. Furthermore, we investigated in how far our thermodynamic model enables to rationalize the observed negative gas adsorption with methane in DUT-49(Cu).

The Co(bdp) (BDP²⁻ = 1,4-benzenedipyrazolate) was originally synthesized by the group of Jeffrey R. Long in 2008.¹ The structure consists of pairs of Co²⁺ ions along a one-dimensional chain, which are tetrahedrally coordinated by N atoms from four independent BDP²⁻ ligands. Two structures were resolved in the original paper: the Co(BDP).2DEF.H₂O (as-synthesized) and the fully desolvated Co(BDP). Some cell parameters and characteristic volumes, as determined experimentally are taken up in Table R1.1 of this review letter. The as-synthesized material was determined to have a tetragonal crystal structure with a cell volume of 2458 Å³. After a full desolvation, the X-ray powder diffraction pattern indicated a complete conversion of the original tetragonal structure to a contracted monoclinic phase with a substantial volume change towards 1183 Å³. In 2010 a spectacular structural response of the material was experimentally evidenced by Salles and co-workers upon adsorption of nonpolar molecules such as N₂ and H₂.² N₂ gas sorption measurements at 77 K revealed a five-step adsorption process, as schematically shown in Figure R1.1. Later in 2015 the response of the material was studied upon methane adsorption by Mason *et al.*³ It was found that at low methane pressures a contracted phase¹ with a volume of 1183 Å³ was observed, whereas at higher methane pressures a large pore phase

¹ We use the terminology contracted phase and large pore phase here for the sake of consistency with the terminology used in the main paper. In the original paper of Mason *et al.* the terminology collapsed phase and expanded phase was used.

was obtained with a volume of 2294 Å³. The unit cell properties of all resolved structures are summarized in Table R1.1 of this response letter together with structural data of the simulated structures performed for this paper.

	method	T (K)	a (Å)	b (Å)	c (Å)	alpha (°)	beta (°)	gamma (°)	Volume (Å ³)
Co(bdp)	PBE+D3(BJ) (this work)	0	24.853	5.907	7.160	90.0	92.8	90.0	1050
Co(bdp)	PBE+D3(BJ) (this work)	0	18.855	18.855	7.134	90.0	90.0	90.0	2536
Co(bdp)	FF (this work)	0	18.878	18.939	7.073	93.2	90.0	90.0	2525
Co(bdp)	FF (this work)	300	18.689	18.648	6.978	89.9	89.8	89.6	2470
Co(bdp)	FF (this work)	300	6.204	24.679	6.933	90.0	90.0	90.0	1060
Co(bdp) + 2xCH ₄	FF (this work)	300	8.947	24.368	6.942	89.9	90.1	90.0	1510
Co(bdp) + 8xCH ₄	FF (this work)	300	18.704	18.751	7.002	90.0	90.1	90.1	2460
Co(bdp)+2DEF+H ₂ O ¹	SCXRD	193	18.742	18.742	6.998	90.0	90.0	90.0	2458
Co(bdp) + 30 bar CH ₄ ³	In situ PXRD	298	21.763	15.220	6.982	90.0	90.0	97.4	2294
Co(bdp) ³	In situ PXRD	298	24.827	7.146	6.675	90.0	90.0	92.6	1183

Table R1.1 : Characteristic unit cell parameters and volumes of Co(bdp), as determined experimentally and theoretically from this work.

Figure R1.1: Simulated structures of Co(BDP), representing pressure-dependent pore evolution via a five-step phase transition. The view down a single channel (left) and a portion of the Co^{II} chain (right) are shown for each structure with Co, N, and C atoms depicted as pink, blue, and gray spheres, respectively. Pressure ranges in which each phase was observed are vacuum, 0.2–0.25 bar, 0.45–0.55 bar, 1.9–3.2 bar, and 7.0–9.5 bar for dry, Int.1, Int.2, Int.3, and filled, respectively. Note that the monoclinic unit cell angle α for dry to Int.3 corresponds to the dihedral angle between two pyrazolate rings. [Reprinted with permission from ². Copyright (2017) American Chemical Society]

We applied the thermodynamic model on both the empty framework and structures loaded with a varying amount of methane molecules. To this end, we first constructed a new force field, with our QuickFF routine following the computational details described in the main manuscript. A series of periodic Density Functional Theory calculations were performed using VASP, on initial structures resembling the contracted pore and large pore phases as reported in reference.² We constructed an equation-of-state around the large pore phase and used these input data for the construction of the force field. The equation-of-state resulting from the DFT calculations at 0 K are shown in Figure R1.2.

Figure R1.2: Equation-of-state obtained from the DFT based calculations at the PBE-D3(BJ) level of theory.

Structural transitions induced by a mechanical pressure were studied by constructing the thermodynamic potential in terms of the mechanical equations of state and the volume versus pressure curves. The results are shown in Figure R1.3. Interestingly two empty phases are found: a large pore phase with a volume of 2470 Å³ and a contracted pore phase with a volume of 1060 Å³. The contracted pore phase is substantially more stable by about 45 kJ/mol at 300 K. This structure resembles the experimentally observed desolvated phase, which is stabilized by π - π interactions between the aryl rings of the neighboring BDP²⁻ ligands. Following the nomenclature introduced in the paper, Co(bdp) belongs to the class of Type II materials and behaves as a triggered disperser (Type IIb). Experimentally this pressure induced behavior of the material was not observed yet, as the current synthesis procedures only obtained the contracted pore phase when fully desolvating the as-synthesized structure. These predictions are expected to boost the experimentalists, to develop another synthesis/activation procedure in order to prepare the material in its large pore phase and to further control its performance in a compression/decompression cycle.

We have included the results of the Co(bdp) in Figure 4(a) and discussed the results on page 6 of the manuscript. The following paragraph was added:

“Similar pressure-induced flexibility is predicted from our model for the Co(bdp) material, not belonging to the MIL-53 series. Current experimental procedures only allowed to synthesize a fully desolvated structure in the contracted pore phase, whereas our model predicts the existence of a large and contracted empty pore phase which is only slightly less stable (45 kJ/mol at 300 K). This material might be interesting for further testing in a compression/decompression cycle”

In a next step, we used the proposed thermodynamic model to study the methane induced flexibility of the Co(bdp) material which was experimentally observed. We adopted the procedures explained in the paper for the case where the trigger $X \neq P$. As a result, a series of $P_T(V, N)$ profiles were constructed with the newly developed force field. The thermodynamic potential in terms of the volume, the mechanical equations of state $P_T(V, N)$ and $V(N)$ are shown in Figure R1.3 for a varying number of molecules per unit cell. Our model predicts that the material undergoes a phase transition from a contracted pore to a large pore phase, when initially starting from a contracted pore phase, for a loading of methane per unit cell varying between 2 and 4 molecules. The characteristics of the unit cell and volume of the various phases are indicated in Table R1.1. For phases derived from the MD simulations at 300 K, the cell parameters were obtained by averaging over the MD ($NV\sigma_\alpha = 0T$) simulations with a volume taken from the grid point which is the closest to the minimum (i.e. with the pressure ≈ 1 bar). Experimentally, a transition from a contracted pore phase to a large pore phase was found for a methane pressure between 16 bar corresponding to 1 CH₄ adsorbed molecule per unit cell volume of 1183 Å³ and 26 bar corresponding to 7 CH₄ adsorbed molecules per unit cell volume of 2305 Å³.³ Our thermodynamic model agrees with the experimentally-observed methane induced flexibility of the Co(bdp).

We incorporated our results of methane adsorption in Co(bdp) in the manuscript. The results are shown in Figure 4 (c), which is repeated here in Figure R1.5. The original results on xenon adsorbed in MIL-53(Al) are still discussed in the main paper but the corresponding figures with the mechanical equations of state, pressure versus external trigger and volume versus number of particles are now shown in the Supplementary Information. As such we meet the comment of the reviewer to incorporate results on more exciting materials of the recent literature.

The text has also slightly been adapted to incorporate the results of Co(bdp) in the main paper as shown below:

“Guest-induced phase transformations are illustrated for xenon adsorption in MIL-53(Al) (Figure S2 of the SI), methane adsorption in Co(bdp) (Figure 4c) and benzene adsorption in DMOF-1(Zn) (Figure 4d). We investigated the response of all these materials upon guest loading by constructing a series of $P_T(V, N)$ mechanical equations of state for a fixed temperature but with a varying number of particles (see Methods section). Experimentally, one controls the chemical potential, however computationally it is easier to perform molecular dynamics simulations by controlling the number of particles. The obtained $P_T(V, N)$ profile can easily be transformed to a $P_T(V, \mu)$ equation of state by means of a

Legendre transformation of the corresponding Helmholtz free energy to the grand canonical potential.⁴ However, since we here investigate the number of particles as a trigger, we proceed by localizing the stable points at the intersection of the pressure profiles with a fixed mechanical pressure of 1 bar. Hence, the $V(N)$ response curves upon guest loading and at fixed mechanical pressure, corresponding to atmospheric pressure, are constructed (Figure 4c for methane@Co(bdp) and Figure S2 for xenon@MIL-53(Al)). For xenon adsorption in MIL-53(Al), our model predicts that the material undergoes a phase transition to an intermediate phase with a volume in between the contracted pore and open pore phases, when initially starting from the open pore phase, for a loading of one xenon particle per unit cell. Upon higher loading the volume gradually increases. These results agree well with the experiments of Boutin et al., which evidenced breathing transitions in the measured xenon adsorption isotherms for MIL-53(Al) in the temperature range 195–323 K⁵ and earlier simulations based on a mean-field model.^{5,6} For Co(bdp) we predict a transition from a contracted to an open pore phase for a methane loading varying between 2-4 molecules, which agrees well with the experiments of Mason et al.³

Figure R1.3 : Illustration of the mechanical equations of state for methane@Co(bdp) at 300 K. Left: Free energy profile; Middle: Mechanical equations of state; Right: Volume versus methane loading.

Finally we also studied the pressure induced behavior of the DUT-49(Cu) as requested by the reviewer. DUT-49(Cu) is composed of copper paddlewheels connected through 9,9'-([1,1'-biphenyl]-4,4'-diyl)bis(9H-carbazole-3,6-dicarboxylate)(BBCDC) ligands and it was originally synthesized by the group of Kaskel in 2012.⁷ In 2016 the same group showed a spectacular negative gas adsorption for methane and *n*-butane.⁸ This negative gas adsorption was associated with a phase transition from the open pore phase to a contracted pore phase, characterized on the microscopic scale by the buckling of the BBCDC ligand. Very recently, Evans and co-workers performed a theoretical study on the DUT-49(Cu) material to rationalize the negative gas adsorption. They investigated the influence of the external pressure and methane adsorption on the flexibility of this material.⁹ To this end, free energy versus volume curves were constructed using the procedure outlined in reference.¹⁰ These simulations were performed in the NVh_0T ensemble where the shape fluctuations are not taken into account.

Starting from the force field reported by Evans *et al.* we studied the mechanical response of the empty DUT-49(Cu) material using our thermodynamic model. Within the time frame of this review, it was impossible to use our own QuickFF routine to derive an in-house developed force field, given the large amount of atoms in the unit cell for which DFT calculations would have to be performed. The empty DUT-49(Cu) framework contains 1728 atoms. Because of symmetry, this number can be reduced by at least factor 4. Therefore, we started from the force field proposed by Evans to study the mechanical induced flexibility which was not yet experimentally measured. The results are shown in Figure R1.4

We find similar results as reported by Evans *et al.* and predict that DUT-49(Cu) can be classified as a **Type IIa** material and is hence a potential candidate as shock absorber. If the material is originally in its large pore phase, it will undergo a phase transition to the contracted pore phase upon external pressures higher than 50 MPa. When releasing the pressure the material would remain in the metastable contracted phase.

Figure R1.4: Free energy profile, mechanical equation of state and volume versus pressure for DUT-49(Cu) at 300 K obtained with the force field developed by Evans *et al.*⁹

We added the pressure-induced-flexibility behavior of DUT-49(Cu) in the main manuscript, in the section where Type IIa materials are discussed on page 6.

“DUT-49(Cu) is another example of a potential shock absorber (Figure 4a).⁷ According to our simulations, DUT-49(Cu) is predicted to have the ability to store more mechanical energy as MIL-53(Al)-F (57.8 J/g vs 7.9 J/g). This might seem contradictory, as the transition pressure of DUT-49(Cu) is lower, however, as explained in the SI, the larger volume variation (per gram material) is the main cause for the higher amount of absorbed mechanical energy.”

Furthermore we incorporated the results of DUT-49(Cu) in Figure 4(a) of the main manuscript as shown below.

Figure R1.5: Updated figure 4 from the main manuscript.

The following section has been added to the Supplementary Information about the mechanical energy stored in MIL-53(Al)-F and DUT-49(Cu):

Approximate calculation of mechanical energy stored in MIL-53(Al)-F and DUT-49(Cu):

An approximate method to calculate the mechanical energy stored in the material during one compression/decompression cycle consists of considering the transition pressure times the volume change associated with the phase transition. This is only an approximate calculation since it assumes that the transition occurs reversibly at a fixed pressure given by the transition pressure. However, during the transition itself, the system is not in mechanical equilibrium with its environment. As a result, the transition is not a quasi-static process, which also makes it irreversible and this method is hence only approximate. Using this method, we find values of 25.6 J/g for MIL-53(Al)-F and 92.3 J/g for DUT-49(Cu). Although the transition pressure is lower in the case of DUT-49(Cu) (60.3 MPa) compared to MIL-53(Al)-F (72.4 MPa), the capacity for storing mechanical energy is larger due to the larger volume change (1.5 cm³/g for DUT-49(Cu) versus 0.4 cm³/g for MIL-53(Al)-F).

Response in how far the negative gas adsorption could be detected by the thermodynamic model.

This is a very interesting point, which warrants a more fundamental thermodynamic discussion. We shortly mentioned the potential extension of the model to explain negative gas adsorption as observed by Kaskel et al. ⁸ in the manuscript on page 10 (vide infra).

The starting point of our thermodynamic model is the Helmholtz free energy $F(N, V, T)$ with state variables temperature, volume and number of particles. However, one can transform one thermodynamic potential to another by means of a Legendre transformation, as noted in the paper on page 3.⁴

In an experimental set-up, one typically controls the mechanical pressure on the MOF, the chemical potential of the adsorbent species in the environment and the temperature. In many cases, those discussed below, the mechanical pressure P is equal to the vapor pressure p_{vap} exerted by the gas. The chemical potential and the vapor pressure are coupled by means of the equation of state. To detect the phenomenon of negative gas adsorption, one needs to construct the adsorption isotherm $N_T(p_{vap})$ which yields the number of adsorbed particles in terms of the vapor pressure. A negative gas adsorption corresponds to a region where the number of particles decreases when increasing the vapor pressure. This phenomenon was experimentally observed for DUT-49(Cu) as shown in Figure R1.6. ⁸

Figure R1.6: Adsorption isotherm for methane in DUT-49(Cu) at 111 K. Reprinted with permission from ⁸ Copyright (2017) Nature Publishing Group.

The thermodynamic model as presented in our paper yields the volume response with respect to an external stimulus (Figure 4 third row) and yields direct information on the framework flexibility in terms of an external stimulus but does not allow to directly detect the phenomenon of negative gas adsorption. To obtain such an information, one needs to transform the $F_T(V; N)$ curves towards the

thermodynamic potential in the osmotic ensemble in terms of the control variables (μ, P, T) by means of a Legendre transformation:

$$X_T(\mu, P) = \min_{(N, V)} \{F_T(V; N) + PV - \mu N\}$$

Such a procedure was already applied by Vanduyfhuys et al.⁴ In the latter paper, the minimization was performed in two steps: first a minimization is performed in terms of the number of particles for fixed values of μ and P :

$$\frac{\partial \tilde{X}_T}{\partial N} = 0$$

imposing chemical equilibrium, where $\tilde{X}_T(N, V; \mu, P) = F + PV - \mu N$ is an intermediate potential in which N and V are treated as independent variables and μ and P are fixed parameters. As a result the thermodynamic potential is obtained $\tilde{X}_T(V; \mu, P)$, in which the chemical potential indeed acts as a trigger, but also the amount of adsorbed guest particles $N_T(V; \mu, P)$. As was mentioned earlier, the mechanical pressure P equals the vapor pressure p_{vap} . Therefore, from now on, we will drop either μ or P from the argument list of \tilde{X}_T and N_T and assume μ to be a function of p_{vap} through the equation of state. Furthermore, the pressure term PV in \tilde{X}_T can be neglected for vapor pressures in the order of 1 bar. In the following, we ignore this term and as a result, the intermediate potential $\tilde{X}_T(V; \mu, P)$ actually becomes the grand canonical potential $F_T(V; \mu)$ in the (μ, V, T) ensemble.

We illustrate the procedure for xenon@MIL-53(Al) at 300 K. Figure R1.7(a) shows the $\tilde{X}_T(V; p_{vap})$, in terms of the volume whereas Figure R1.7(b) shows the number of adsorbed guest molecules in terms of the volume for various values of the vapor pressure. In principle one needs to perform the second minimization step in terms of the volume on this thermodynamic potential. By applying a same procedure as explained in Figure 5 of the manuscript on $\tilde{X}_T(V; \mu)$, but with a mechanical pressure given by the vapor pressure $P_0 = p_{vap}(\mu)$ instead of the atmospheric pressure $P_0 = 1$ bar, one obtains the volume response with respect to the chemical potential (or vapor pressure), which is visualized for the case of xenon@MIL-53(Al) in Figure R1.8(a).

Figure R1.7: Results for applying the first Legendre transform of number of particles N to chemical potential $\mu = \mu(p_{vap})$ for Xe in MIL-53(Al). (a) thermodynamic potential $\tilde{X}_T(V; p_{vap})$ and (b) number of particles $N_T(V; p_{vap})$ for various values of the vapor pressure p_{vap} .

Figure R1.8: Results for applying the second Legendre transform of volume V to mechanical pressure $P = p_{vap}$ for Xe in MIL-53(Al). (a) volume response to vapor pressure (b) adsorption isotherm. Blue squares represent large pore phase, red circles represent contracted pore phase.

We now rationalize how the profiles of $\tilde{X}_T(V; p_{vap})$ and $N_T(V; p_{vap})$ look in case of positive and negative gas adsorption. Suppose we start for the xenon@MIL-53(Al) initially in the large pore phase. This corresponds to the empty phase most encountered experimentally, since the material is activated at high temperature. When increasing the vapor pressure, a new minimum appears corresponding to an intermediate phase. The transition from the large pore phase to the intermediate phase only occurs when the large pore minimum in the osmotic potential disappears. Further increasing the vapor pressure finally yields back the large pore phase. In this case positive gas adsorption is observed, as can be detected by inspecting the $N_T(V; p_{vap})$ profiles. As the vapor pressure increases, the structure transforms at a vapor pressure of 0.16 bar (see arrow for the profile at 0.16 bar in Figure R1.7). At this transition, the number of particles increases in case of the xenon@MIL-53(Al). To truly compare with experiment one can perform the second minimization step of the Legendre transformation and construct the adsorption isotherm. The results of this second minimization step for xenon@MIL-53(Al) are shown in Figure R1.8(b).

To test the hypothesis further we started from the free energy profiles at 120 K published by Evans et al.⁹(see Figure R1.9).

Figure R1.9: $F_T(V; N)$ profiles computed by Evans et al. for DUT-49(Cu) at 120 K
Figure reproduced with the data provided by Evans et al.⁹

Following the procedure with the Legendre transform sketched above, Figure R1.10 shows the results of applying the first Legendre transform to these profiles, i.e. the thermodynamic potential $\tilde{X}_T(V; p_{vap})$ as well as the number of particles $N_T(V; p_{vap})$ for various values of the vapor pressure p_{vap} of methane. From these results, one would also conclude that no structural transition is observed when collective behavior is assumed and the system always remains in the large pore phase, because this large pore minimum never disappears. However, it could be that the transition is not observed due to missing data in the original $F_T(V; N)$ profiles. Looking at the $F_T(V; N)$ in Figure R1.9, extra $F_T(V; N)$ curves for a number of methane molecules between 400 and 600 could prove crucial, because, as Evans et al. mentioned, that is exactly the range in which the barrier between large pore and contracted pore decreases and hence where transitions could happen. From Figure R1.10(a), we could hypothesize, however, that if a structural transition would occur, it will be at a vapor pressure of around 7.4 bar (see arrow in Figure R1.10), because this corresponds to the thermodynamic potential with the lowest barrier. Furthermore, if we now consider Figure R1.10(b), we observe that at 7.4 bar, the large pore already contains more methane than the contracted pore, so if the transition from large pore to contracted pore indeed occurs, the amount of adsorbed methane would decrease. In other words, negative gas adsorption would occur.

Figure R1.10: Results for applying the first Legendre transform of number of particles N to chemical potential $\mu = \mu(p_{vap})$ for CH_4 in DUT-49(Cu). (a) thermodynamic potential $\tilde{X}_T(V; p_{vap})$ and (b) number of particles $N_T(V; p_{vap})$ for various values of the vapor pressure p_{vap} . The $F_T(V; N)$ curves required for the Legendre transform were provided by Evans et al. ⁹

To further illustrate this, Figure R1.11(a) also shows the response of the cell volume of DUT-49(Cu) to the vapor pressure of methane and Figure R1.11(b) shows the adsorption isotherm of methane. As we can see in the adsorption isotherm, a strong increase in the amount of adsorbed methane for the large pore (blue squares) occurs at around 6 bar, and as a result the amount of adsorbed methane in the large pore (blue squares) becomes larger than in the narrow pore (red circles). Hence, if the transition occurs at pressures higher than 6 bar, negative gas adsorption would indeed occur.

Figure R1.11: Results for applying the second Legendre transform of volume V to mechanical pressure $P = p_{vap}$ for CH_4 in DUT-49(Cu). (a) volume response to vapor pressure (b) adsorption isotherm. Blue squares represent large pore, red circles represent narrow pore.

"In principle our thermodynamic model yields all necessary components to also explain unexpected effects such as negative gas adsorption, which was observed for DUT-49(Cu) upon exposure of methane, as explained in section 5 of the SI."

Minor points:

Is it possible to investigate the role of entropy more explicitly? Particularly, the temperature driven flexibility is determined by the balance of dispersion interactions and lattice entropic gain, hence a study of entropy would be very interesting at least for $X = T$.

The temperature driven flexibility is indeed believed to be governed by a subtle balance between long-range dispersion interactions in the framework and entropy contributions at finite temperature.¹¹ A proper treatment of entropy for flexible materials belongs still to one of the most challenging problems of materials modelling. It requires accurate sampling of the potential energy surface and methods which go beyond the harmonic oscillator approximation.¹²⁻¹⁴ From the free energy estimation methods used in this paper, we can deduce the entropy contribution at a given temperature T in terms of the volume. Although it is indeed very challenging to compute the accurate entropy differences in terms of different temperatures $\Delta_T S = S(T_2) - S(T_1)$, we can, however, extract entropy differences in terms of different volumes $\Delta_V S = S(V_2) - S(V_1)$ by computing the free energy difference as well as the difference in internal energy and applying the formula $\Delta_V S = \frac{\Delta_V E - \Delta_V F}{T}$. Furthermore, we can compute $\Delta_V S$ for various temperatures to investigate the influence of temperature on entropy differences. Note that this procedure only allows us to compute $\Delta_T \Delta_V S$ and not $\Delta_T S$.

We studied more in depth how the entropy varies in terms of the volume and for various temperatures for MIL-53(Al)-F and for xenon@Mil-53(Al). Such an analysis should allow to give more insight into the larger contribution of entropy to the large pore phase, which is thought to be governing the phase transformation. Figure R1.12 shows the decomposition of the free energy of the MIL-53(Al)-F into the internal energy and the entropy contribution for the force-field based MD simulations at various temperatures. Note that the reference level for both the internal energy and entropy is arbitrarily chosen to enable a visualization of all curves on the same plot. Hence only energy/entropy differences between different volumes $\Delta_V S$ are meaningful. As mentioned before we are unable to deduce the entropy explicit in terms of temperature.¹⁴

(a) Pressure

(b) Internal energy

(c) Free energy

(d) Entropy

Figure R1.12: (a) Pressure as well as (b) internal energy as function of volume from force-field based MD simulations in the $(NV\sigma_\alpha = 0T)$ ensemble for the MIL-53(Al)-F. From these profiles, (c) the free energy and (d) entropy profiles are derived using thermodynamic integration and the relation $S = \frac{E-F}{T}$ respectively.

The internal energy is almost independent of the temperature (around 2 kJ/mol variation on the large pore-contracted pore stability). Our simulations using classical molecular dynamics indicate that the entropy variation in terms of the volume is nearly independent on the temperature. Such effects were

already previously observed in other systems.¹⁵ However, further investigation is warranted on this topic, as for instance nuclear quantum effects may play an important role to determine the entropy more accurately. Current results show that the entropy for the large pore phase is substantially higher than for the contracted pore phase ($S(V_{lp}, T) - S(V_{cp}, T) \approx 33 \text{ J}/(\text{mol}\cdot\text{K})$) for a large pore volume V_{lp} and a contracted pore volume V_{cp} taken as the 300 K equilibria). At higher temperatures, the large pore phase is indeed systematically more stable compared to the contracted pore phase, since the entropy contribution $-T(S_{lp} - S_{cp})$ to the free energy becomes more dominant due to its linear dependency on the temperature and yields a systematically larger contribution to the overall free energy. Further investigation would be required to study in detail the temperature independence in terms of the volume but this is beyond the scope of the current paper.

A similar analysis was performed for the xenon@MIL-53(Al). A decomposition of the free energy into the internal energy and entropy is performed for various xenon loadings. In this case the internal energy does depend on the xenon concentration, which is related to the contributions from the xenon-host and xenon-xenon interactions. Also the entropy is largely dependent on the number of particles, as expected since higher loadings mean less free space and hence lower entropy.

We added in the methods section the following sentence and reference to the Supplementary Information:

From the output of such molecular dynamics simulations, the time average of the instantaneous hydrostatic pressure P_i is computed, which is defined at each time step as $P_i = \text{Tr}(\sigma_i)/3$ with σ_i the instantaneous, internal stress tensor resulting in $P(V)$ and $F(V)$ profiles. More information about the procedure can be found in the work of Rogge et al.¹⁰ For a selected set of materials, the separate contribution of internal energy and entropy was deduced from the simulations to obtain more insight into the effects contributing to the stabilization of one or the other phase (Section 6 of the SI).

Figure1: for a non-expert in the field, the inorganic building unit shown in Figure 1 is somewhat confusing. I suggest updating the figures, in particular the metal nodes and include the coordination environment of the metals.

We updated the figure as requested by the reviewer 1. Furthermore, the figure was extended to include also Co(bdp) and DUT-49(Cu) as these material is now discussed in the revised version. The new figure is shown below.

Figure R1.13: Illustration of some metal-organic frameworks which show potential stimuli-responsive behavior. All shown materials are MOFs built from a 0D or 1D inorganic moieties connected by organic linkers.

Page 2, line 3 “techniques such as mercury-intrusion porosimetry, high-pressure X-ray diffraction, differential scanning calorimetry or inelastic neutron scattering”. Either add Terahertz spectroscopy at the end, or merge it together with inelastic neutron scattering to spectroscopic techniques. To my knowledge, yet there are no temperature/pressure dependent neutron scattering studies on flexible MOFs.

The sentence has been adapted as requested by the reviewer.

“Experimentally, framework flexibility can be followed by monitoring the response of the material to the external stimulus (schematically shown in **Figure 2**) with techniques such as mercury-intrusion porosimetry¹⁶, high-pressure X-ray diffraction¹⁷, differential scanning calorimetry¹⁸ or spectroscopic techniques.¹⁹”

Typo at page 6, very top “remain remain”.

All minor corrections have been adapted.

Reply to Reviewer #2

We thank the reviewer for his/her constructive comments, which helped to improve the manuscript to a great extent. We have taken all comments of the reviewer into account and adapted the manuscript accordingly. Below the comments of the reviewer are copied (in italics) with an answer from our side.

The authors demonstrated the prediction and calculation of how "soft" behaviors of metal organic framework (MOF) crystals are. As a background, it is well known that a certain amount of MOF crystal show dynamic/flexible structural transformation upon external stimuli or guest accommodation. Many works are reported for each compound, and it is desired to generalize the flexible behavior for MOFs by the theoretical approach by use of force fields and other techniques with using of crystal structure. Because the approach could elucidate more useful insights of the flexible nature for other MOFs for future-application as the author mentioned in Figure 3 and the section of conclusion.

I read the manuscript carefully, and found the proposed results and discussion is of significant to understand the unique nature of soft MOF and realized that the approach is powerful to illustrate the different types of flexible behaviors in various MOFs. I found several new information which are not evaluated by the experiments, and the results herein support the experiments and prediction of the flexibility-based functionality of MOFs. The paper mostly focuses on the three-dimensional structured MOFs but there are many flexible MOFs with two-dimensional or even one-dimensional coordination polymers (e.g. Nano Lett 2006, 6, 2581-2584.). I am not sure how the present approach could be available for the other flexible MOFs/CPs in terms of creation of force field, and the authors should add explanation about this point. I also wondered how the behavior of crystal-to-amorphous (reversible) transformation which is recently highlighted will be explained based on the proposed approach (e.g. Acc. Chem. Res. 2014, 47, 1555-1562. or APL Materials 2014, 2, 124401.). The authors only studied on the crystalline nature but the expansion of the discussion also for glass or amorphous MOFs would become more strong to generalize the work in here. At least the authors should mention/discuss the potential or perspective to access this issue in this manuscript.

The comments raised by the reviewer towards the extension to two-dimensional or one-dimensional coordination polymers, crystal-to-amorphous transformation or amorphous MOFs is particularly interesting. The proposed thermodynamic model is in principle broadly applicable, only a few assumptions are made. First the volume needs to represent a representative collective variable which enables to detect the breathing flexibility investigated here (as mentioned on Page 1 of the paper). Second, collective behavior is assumed as mentioned on Page 8 of the paper [A transition to the contracted pore phase is predicted to be only possible if the open pore minimum in the free energy profile would disappear, according to the assumption of collective behavior.⁴⁰]

Application towards 1D and 2D coordination polymers

While our model is conceptually applicable for 1D or 2D coordination polymers, deriving reliable force fields for these type of materials is not a straightforward task. For the mentioned materials, the structure is very much dominated by non-bonding interactions between 1D chains or between the layers. For the molecular framework Ag(tcm) (tcm= tricyanomethanide)²⁰ with a layer-like topology, we derived a force field based on Density Functional Theory based input. However after optimization of the structure with the new force field, the layers drifted apart, which is an indication that the non-bonding interactions from the MM3 force field, which are used most commonly complementary to the covalent force field derived by QuickFF, are not fully suited. Such a study on the non-bonding interactions in lower dimensional coordination polymers is beyond the scope of the current paper. Indeed, as mentioned in literature, non-bonding interactions can only be described well by expensive high-level calculations in both 1D²¹ and layered 2D systems.²²

Application towards amorphous materials

Soft porous materials lacking long-range order or soft porous materials for which the phase transition is accompanied by a loss of long-range order cannot be directly simulated with the protocol suggested here, as our procedure relies on periodic boundary conditions. Using these periodic boundary conditions, the length scale on which the loss of order can be described is limited to the size of the unit cell, such that only short-range loss of order can be described within our model. In contrast, amorphous metal-organic frameworks (aMOFs) retain the basic building blocks and connectivity of their crystalline counterparts, and are hence highly ordered on these short length scales, but lack any long-range periodic order, as mentioned in the review of Bennett *et al.*²³ To date, simulation of such systems hence remains very challenging with atomistic based models, as constructing sufficiently large unit cells of these materials is computationally restrictively expensive. Hence, to describe these long-range effects, one needs to rely on an approximate description of the system with fewer degrees of freedom, such as the simplified Hamiltonian to describe the adsorption-induced layer-by-layer LP-to-CP transition in MIL-53²⁴, or coarse-grained models, which have already proven to be applicable for the description of biomolecules,²⁵ but are only slowly being introduced for framework materials.²⁶

Following paragraph has been added to the main manuscript at page 4 and 10:

The approach is generic and may be applied **directly** to any soft porous crystal exhibiting long-range order. **Materials for which the phase transition is accompanied by loss of long-range order, such as amorphous MOFs,²³ cannot be straightforwardly simulated as the present approach relies on periodic boundary conditions. Effects such as amorphization under elevated pressures may however be detected both experimentally and theoretically by a broadening of the peaks in the radial distribution function.**²⁷

Our thermodynamic model is in principle generic but relies on reliable force fields. As a typical illustration, we compared the results obtained for DMOF-1(Zn) with various force fields (section 9.1 of the SI). For some materials such as two-dimensional polymers, force field development might be very

challenging due to critical dependence on the non-bonding interactions between the layers or the chains.^{21,22}

This work has enough generality and novelty to explain/predict the flexible nature of various MOFs and it would contribute to explore the new functions and application in the future which would be worth to be published in this journal, but I address some points to be revised to consider the final decision as following.

1. The equation in page 2 (last part) should have an explanation for the term of $\sum \mu_i dN_i$.

If we start from the well-known fundamental relation of thermodynamics for closed systems $dE = TdS - PdV$, we can extend this equation to account for adsorption of guest molecules in open systems. For each species i with N_i particles and chemical potential μ_i , an extra term $\mu_i dN_i$ can be included in the fundamental relation, which accounts for the work needed to adsorb particles in the framework. As such, we arrive at

$$dE = TdS - PdV + \sum_i \mu_i dN_i$$

Finally, by using the definition of the Helmholtz free energy $F = E - TS$, we arrive at:

$$dF = -SdT - PdV + \sum_i \mu_i dN_i$$

2. Figure 2 has many MOF names but many of them are not dealt with in this paper. NOTT-300, Zn(CN)₂, STA-12(Co) etc. The information confuses the readers and I recommend them to be moved to the supporting information or delete. Also there are many abbreviation of MOFs and it is better to describe the precise formula of each MOF in the main manuscript, otherwise it is not friendly to the non-MOF scientists. Figure 1 is not sufficient to let people understand the metal-ligand connectivity and assembled open structure, DMOF-1(Zn) has two ligands but the current Figure 1 does not have such information.

As this is a paper intended for a general audience, we agree with the reviewer that MOF-specific terminology should be introduced appropriately. However, we opted not to remove all materials from Figure 2, as they give an indication that the represented phenomena were found in various materials. Instead, we have now included a table in the Supplementary Information (as shown below) yielding the precise formula for each material mentioned in the paper.

Table R2.1: Name and chemical formula of the MOFs mentioned in this work.

MOF Name	chemical formula
Al-fum	Al(OH)(fumarate)
CAU-13	Al(OH)(trans-cdc)
MIL-47(V)	V ^{IV} (O)(bdc)
MIL-53(Al,Ga,Cr)	(Al,Ga,Cr)(OH)(bdc)
NOTT-300	Al ₂ (OH) ₂ (bptc)
CPL-2	Cu ₂ (pzdc) ₂ (bpy)
DMOF-1(Zn)	Zn ₂ (bdc) ₂ (dabco)
STA-12(Co)	Co ₂ (H ₂ O) ₂ L
Zn(CN) ₂	Zn(CN) ₂
Co ₃ (OH) ₂ (btca) ₂	Co ₃ (OH) ₂ (btca) ₂
Co(bdp)	Co(bdp)
Sc ₂ (bdc) ₃	Sc ₂ (bdc) ₃

Glossary

bdc = 1,4-benzenedicarboxylate	pzdc = pyrazine-2,3-dicarboxylate
cdc = 1,4-cyclohexanedicarboxylate	bpy = 4,4'-bipyridyl
bptc = biphenyl-3,3',5,5'-tetracarboxylate	dabco = 1,4-diazabicyclo[2,2,2]octane
bdp = 1,4-benzenedipyrizolate	btca = benzotriazide-5-carboxylate
L = O ₃ PCH ₂ NC ₄ H ₈ NCH ₂ PO ₃ = N,N'-piperazinebis(methylenephosphonate)	

Figure 1 has been adapted (as shown below) according to the remarks of Reviewer 1 and 2. It now clearly shows the metal-ligand connectivity and the proper ligands for DMOF-1(Zn).

Figure R2.1: Illustration of some metal-organic frameworks which show potential stimuli-responsive behavior. All shown materials are MOFs built from a 0D or 1D inorganic moieties connected by organic linkers.

3. In principle they employed simple MOFs which means the structures have high symmetry, simple composition, not large cell. On the other hand, there are many important MOFs having low symmetry, large unit cell, and often various organic ligands are involved in one open structures (e.g. *Angew. Chem. Int. Ed.* 2010, 49, 4820-4824.). How about the (potential) solution to understand the flexible nature by the proposed method? In recent, mixed ligand approach is specifically important to tune the flexibility for application. Please provide some additional data or at least discussion.

While MOFs with a larger unit cell and a lower symmetry require more CPU time to reach convergence, our protocol does not directly depend on the size of the unit cell nor the symmetry of the material. As a proof of this statement and as requested by Reviewer 1, we included DUT-49(Cu) as a more challenging material with a larger unit cell (1728 atoms in the empty unit cell). However, our protocol requires the volume to be a good collective variable, distinguishing between the different (meta)stable states such as the large pore and contracted pore phases. Hence, one has to be careful to identify all metastable states at the given thermodynamic conditions. Whereas high symmetric materials often have only one or two (meta)stable states as a function of the volume, less symmetric materials may exhibit multiple (meta)stable states which are degenerate, which makes it more difficult to identify all (meta)stable states. In that case, one has to ensure that all (meta)stable states are sampled to obtain reliable pressure and free energy profiles. The materials from the paper proposed by the reviewer, are indeed very

challenging to model, both from the perspective of force-field development (van der Waals interactions will be of crucial importance to accurately describe the interdigitated nature of the 2D layers) as well as regarding to the choice of the collective variable (the volume is not a good collective variable for gate-opening).

4. References are required for the parts "...contracted phase occurs at high transition pressure of about 120 MPa and returns back to the open phase upon releasing the pressure below 20 MPa." (page 6) or "...which was found to be negative for MIL-53(Al)-F in contrast to its OH analogue." (page 8).

We have inserted the appropriate references in the text.

If the authors response these points and provide appropriate revision or additional results, I consider the manuscript is acceptable for this journal.

Reply to Reviewer #3

We thank the reviewer for his/her constructive comments, which helped to improve the manuscript to a great extent. We have taken all comments of the reviewer into account and adapted the manuscript accordingly. Below the comments of the reviewer are copied (in italics) with an answer from our side.

The manuscript reports on a systematic analysis of the thermodynamic behavior of flexible porous materials, in particular metal-organic frameworks, obtained through theoretical calculations based on molecular dynamics simulations.

The manuscript is well written and well organized.

The work that has been carried out leads to significant results of general and wide interest to the scientific community. I believe the paper is worthy to be published on Nature Communications but some revision is needed.

I have a few concerns about the theoretical calculations.

1) Authors used a force-field (FF) that has been developed specifically for treating flexible MOFs. As demonstrated by published articles, the FF has been carefully tested, but I wonder how results are affected by the functional form of the FF. Could they try to repeat some of the calculations with other "standard" FFs and see how they work with respect to the QuickFF.

The sensitivity of the results with respect to the applied force field warrants indeed some attention. We choose to investigate the flexibility behavior of DMOF-1(Zn) more in depth to respond to the question of the reviewer. For this material, we used in the submitted paper the force field of Grosch *et al.*²⁸ In this force field, the covalent terms that describe interactions within the BDC and DABCO ligands are taken from the Generalized Amber Force Field (GAFF).²⁹ Covalent interactions involving the metal unit are parametrized using DFT calculations with the M06-2X functional on framework fragments.³⁰ Electrostatic interactions are described using point charges, which are obtained by fitting to the *ab initio* electrostatic potential (from the DFT framework-fragment calculations) using the CHELPG method.³¹ The van der Waals interactions are modeled using a Lennard-Jones potential using the corresponding GAFF parameters.

We now also generated a QuickFF force field for the DMOF-1(Zn) framework, starting from an *ab initio* determined Hessian using periodic plane waves DFT calculations. Similar to the other materials in this study, we employed the PBE exchange-correlation functional together with DFT-D3(BJ) dispersion corrections. More details can be found in the section 1.1 of the SI. Furthermore we also applied a general purpose force field namely UFF4MOF.³² UFF4MOF was developed as an extension of UFF towards MOFs, with new parameters specifically valid for MOF materials. UFF4MOF has been shown very accurate in reproducing unit cell dimensions of a series of MOFs, however it needs to be tested in how far the parameter set is accurate enough to simulate physical phenomena that are more sensitive to the specific shape of the potential energy surface and temperature corrections such as breathing. A recent assessment of various force fields to predict bulk material's properties like the bulk modulus and linear

thermal expansion for a series of MOFs was performed by Boyd et al.³³ Their study only included rigid materials like IRMOF-1, IRMOF-10, HKUST-1 and UiO-66. Whereas the bulk properties were recently predicted, properties which are sensitive to the framework vibrational modes are prone to larger deviations.

For the three tested force fields the results of the thermodynamic model are given in Figure R3.1. All force fields predict that the volume decreases as the benzene loading increases. The results of the QuickFF and the Grosch force field are the closest together. The UFF4MOF force field predicts substantially different results than the other two force fields. UFF4MOF predicts a second branch in the $V(N)$ profile, which is the most stable as soon as two or more benzene molecules are present per unit cell. This branch shows volumes around 1900 \AA^3 which is not in agreement with experiment.

The results of QuickFF and the Grosch force field agree within acceptable limits. The QuickFF force field predicts in general larger volumes than the other force fields as well as experiment. Note however that for zero loading it is close to the *ab initio* volume to which it was fitted. Grosch *et al.* shows a more pronounced decrease in volume upon adsorption of 2 benzene molecules per unit cell. This is accompanied by a transition from a square to a rectangular cell. For QuickFF, the cell remains (more or less) square.

A more thorough investigation on the sensitivity of the results with respect to various force fields would have to be done to generalize the conclusions taken here. However, from the current analysis, it is not advised to use general purpose force fields like UFF4MOF to describe the flexibility induced behavior of soft porous materials. In literature, also MOF-FF - also a first-principles derived force field - was used to generate free energy profiles of the DUT-49(Cu) material in the work of Evans et al.⁹ The results obtained in that work gave reasonable agreement with experiments. MOF-FF was developed by the group of Schmid *et al.* and was designed specifically to describe Metal-organic frameworks.^{34,35} The force field energy expression is very similar to the MM3 expression, the electrostatic interactions are described by Gaussian charges and the covalent parameters are fitted to *ab initio* cluster data using a genetic algorithm. The MOF-FF concept assumes also a building block approach and is in this sense similar to the first generation force fields developed with QuickFF.³⁶ However current QuickFF routines enable to generate the input data from periodic Density Functional Theory calculations as it was performed in this paper.

We added the part on the force field testing in the Supplementary Information (section 9.1) of the paper and made a critical note on the force field sensitivity in the main paper in the Methods section.

"The thermodynamic model is in principle generic but relies on reliable force fields. For DMOF-1(Zn) we compared the results obtained with various force fields (section 9.1 of the SI). For some materials such as two-dimensional polymers, force field development might be very challenging due to critical dependence on the non-bonding interactions between the layers or the chains.^{21,22}"

Figure R3.1: Illustration of the mechanical equations of state for benzene@DMOF-1(Zn) generated with three different force fields (Panel a : with the force field of Grosch et al.²⁸, Panel b: with an in-house developed force field generated by the QuickFF routine,³⁶ Panel c : with a general purpose force field UFF4MOF.³² Upper row : Free energy profile; Middle row : Mechanical equations of state; Bottom row: Volume versus external trigger.

2) *Prediction of thermodynamic properties through molecular dynamics requires rather long simulations to be sure to have a reliable sampling of the PES. According to the supplementary information, authors run simulations for a given time. How can they be sure that the simulation time was long enough? Could they repeat the simulations for different time scales at least for one system?*

To study the convergence of our simulations as a function of the simulation time, we have carried out longer simulations for MIL-53(Al)-F at 100 K, 300 K and 600 K, extending the simulation up to 5 ns compared to the previously reported 1.15 ns simulations while keeping the equilibration time of 100 ps constant. We have chosen this material for this in-depth study such that we can also investigate the effect of temperature on the convergence of the simulations.

For these three temperatures, the root-mean-square deviation (RMSD) was calculated for three key properties, using the results from the 5 ns simulation as the reference: (i) the pressure profile before fitting, *i.e.* containing the raw data obtained from the MD simulation, (ii) the pressure profile after fitting, and (iii) the free energy profile obtained by thermodynamic integration of the pressure profile. The resulting RMSDs are reported in Tables R3.1 (100 K), R3.2 (300 K), and R3.3 (600 K).

Across all temperatures, we observe an RMSD of 3-6 MPa on the pressure before fitting when comparing the 1 ns simulation (indicated in gray, close to the 1.15 ns simulations reported in the main manuscript) with the longer 5 ns simulation (assumed to be converged), which slightly increases with increasing temperature (3.5 MPa at 100 K versus 5.2 MPa at 600 K). This may be attributed to the size of the available phase space, which increases at higher temperatures, whereas the slower exploration of the phase space at 100 K, due to the lower velocities, does not lead to a worse sampling. By fitting the pressure, the RMSD drops to about 1-2 MPa for all temperatures, well within the accuracy one can expect to obtain with this method. This RMSD in the pressure results in an RMSD in the free energy of about 0.1 kJ/mol when comparing the 1 ns simulation with the 5 ns simulation.

To get more insight in which volume regions of the pressure and free energy profiles are most prone to these small imprecisions, we report in Figure R3.1 the pressure profile before fitting (pane a) and the free energy profile obtained by thermodynamic integration (pane c) as a function of the unit cell volume for the six different simulation times at 300 K. Since the different profiles are virtually indistinguishable on this scale, panes (b) and (d) report the difference in pressure, respectively free energy, with respect to the 5 ns simulation. We observe that the deviations decrease with increasing simulation time.

Finally, in Tables R3.4 (100 K), R3.5 (300 K), and R3.6 (600 K), the volumes of the metastable CP and LP states as well as the LP-to-CP and CP-to-LP transition pressures extracted from the pressure profiles with different simulation times are reported. We observe that the deviations due to limited simulation time do only affect these properties to a minor extent. In conclusion, the earlier reported 1.15 ns simulations have converged sufficiently to yield accurate results within a few MPa and a few tenths kJ/mol for the pressure and free energy profiles, respectively, yielding volumes for the metastable states which are accurate within a few Å³.

For DMOF-1(Zn) loaded with benzene molecules and for MIL-53 loaded with xenon atoms, the simulations were also extended and no noticeable differences were observed in the pressure and free energy profiles.

The results of this analysis are taken up in the Supplementary Information section 9.2.

Table R3.1: Root-mean-square deviation (RMSD) of the pressure profile (both before and after fitting) and free energy profile for MIL-53(Al)-F at 100 K, using the 5 ns simulations as a reference. Values reported in the main text were obtained for a simulation run of 1.15 ns.

	0.5 ns	1 ns	2 ns	3 ns	4 ns
Pressure RMSD before fit [MPa]	5.58	3.49	1.55	1.13	0.96
Pressure RMSD after fit [MPa]	3.46	2.06	0.63	0.36	0.33
Free energy RMSD [kJ/mol]	0.23	0.10	0.02	0.02	0.02

Table R3.2: Root-mean-square deviation (RMSD) of the pressure profile (both before and after fitting) and free energy profile for MIL-53(Al)-F at 300 K, using the 5 ns simulations as a reference. Values reported in the main text were obtained for a simulation run of 1.15 ns.

	0.5 ns	1 ns	2 ns	3 ns	4 ns
Pressure RMSD before fit [MPa]	5.86	4.06	2.64	1.79	1.10
Pressure RMSD after fit [MPa]	1.88	1.03	1.16	0.99	0.51
Free energy RMSD [kJ/mol]	0.12	0.05	0.02	0.02	0.05

Table R3.3: Root-mean-square deviation (RMSD) of the pressure profile (both before and after fitting) and free energy profile for MIL-53(Al)-F at 600 K, using the 5 ns simulations as a reference. Values reported in the main text were obtained for a simulation run of 1.15 ns.

	0.5 ns	1 ns	2 ns	3 ns	4 ns
Pressure RMSD before fit [MPa]	8.30	5.23	2.94	2.03	1.25
Pressure RMSD after fit [MPa]	2.39	1.28	0.77	0.64	0.45
Free energy RMSD [kJ/mol]	0.10	0.07	0.06	0.06	0.04

Table R3.4: Volumes of the metastable CP and LP states and the two transition pressures for MIL-53(Al)-F at 100 K as extracted from MD simulations with different simulation times (0.5 - 5 ns). Values reported in the main text were obtained for a simulation run of 1.15 ns.

	0.5 ns	1 ns	2 ns	3 ns	4 ns	5 ns
CP volume [Å³]	872	870	871	870	870	870
LP volume [Å³]	1471	1471	1471	1471	1471	1471
LP-to-CP transition pressure [MPa]	87.4	87.1	87.1	87.1	87.0	87.1
CP-to-LP transition pressure [MPa]	-81.6	-81.2	-81.4	-80.7	-80.7	-80.9

Table R3.5: Volumes of the metastable CP and LP states and the two transition pressures for MIL-53(Al)-F at 300 K as extracted from MD simulations with different simulation times (0.5 - 5 ns). Values reported in the main text were obtained for a simulation run of 1.15 ns.

	0.5 ns	1 ns	2 ns	3 ns	4 ns	5 ns
CP volume [Å³]	914	915	914	914	913	913
LP volume [Å³]	1460	1459	1459	1459	1459	1459
LP-to-CP transition pressure [MPa]	77.8	77.5	77.2	77.2	77.1	77.3
CP-to-LP transition pressure [MPa]	-30.0	-31.2	-31.4	-31.0	-31.4	-31.2

Table R3.6: Volumes of the metastable CP and LP states and the two transition pressures for MIL-53(Al)-F at 600 K as extracted from MD simulations with different simulation times (0.5 - 5 ns). Values reported in the main text were obtained for a simulation run of 1.15 ns.

	0.5 ns	1 ns	2 ns	3 ns	4 ns	5 ns
CP volume [\AA^3]	1020	1020	1020	1020	1020	1020
LP volume [\AA^3]	1441	1441	1441	1441	1441	1441
LP-to-CP transition pressure [MPa]	70.3	69.4	68.6	68.3	67.9	67.8
CP-to-LP transition pressure [MPa]	27.4	29.3	29.4	29.1	29.1	28.8

Figure R3.1: Obtained pressure and free energy profiles for MIL-53(Al)-F at 300 K with different simulation times: (a) Pressure profile before fitting, (b) Difference in pressure profiles using the 5 ns simulation as reference, (c) Free energy profile, (d) Difference in free energy profiles using the 5 ns simulations as reference.

3) Authors adopts a molecular dynamics (MD) approach to predict the thermodynamic behavior of the flexible systems, but they do not mention that the same information could be obtained through lattice dynamics (LD) in a quasi-harmonic approximation, for instance. To confirm their findings and their wider reliability, authors should also try to use LD. In that respect, LD is more suitable to *ab initio* quantum mechanical methods and not only FF. So, this could be an even stronger evidence of the results obtained with classical simulations.

As requested by the reviewer, we also calculated the vibrational entropy via lattice dynamics calculations. More information on the procedure may be found in the reviews of Butler *et al.* and Tkatchenko *et al.*^{12,13} This approach is critically dependent on the accurate calculation of the Hessian matrix, which is called the dynamical matrix for periodic systems. It contains the second order derivatives with respect to the geometry of the system. As a result one obtains a series of phonon modes and phonon eigenvectors. To properly include all non-negligible lattice modes a supercell approach might be necessary, as it is important not to introduce artifacts between periodic images.

To obtain thermal corrections at various volume points we applied the quasi-harmonic approximation, where a series of structures are optimized for a number of volume points. The phonons are still harmonic in this case but thermal corrections which are dependent on the volume are taken into account as the phonons become volume dependent.

We applied this procedure both for MIL-53(Al) and xenon@MIL-53(Al) systems using force fields. In case of force field calculations, the Hessian and forces can be calculated in an analytical way and is thus less prone to numerical uncertainty. We used a 1x2x1 unit cell which is necessary to describe all potential energy terms correctly in the force field expression. Calculations on a 2x4x2 unit cell provide very similar results, from which we conclude that the 1x2x1 supercell is sufficiently large. We choose a grid spacing of 25 Å³. The results are shown below.

Figure R3.2: Energy profiles for MIL-53(Al) computed with both a harmonic and an anharmonic force field. The blue curves (OPT) represent energies of optimized structures without thermal corrections. The orange (QHA) and pink (MD) curves show the free energy computed within the quasi-harmonic approximation and with thermodynamic integration respectively.

For MIL-53(Al) we find that the quasi-harmonic approximation yields fairly good agreement with the MD based results, at least if one uses a force field in which the bonds and bends are described using harmonic terms (which is the case in the submitted version of the paper). To investigate in how far this influences the obtained profiles, we also constructed similar profiles including anharmonic contributions, by adding anharmonic corrections from the MM3 force field to the bond and bend terms.³⁷ It is clear that the quasi-harmonic profile is in this case in worse agreement with the profile obtained from MD simulations, as only in the latter approach it is possible to correctly sample anharmonicities of the potential energy surface.

For the xenon@MIL-53(Al) system the quasi-harmonic oscillator approximation fails to describe the free energy profile accurately. This can be ascribed to two effects. First of all, it is much more difficult to get smooth normal mode frequencies as function of volume because the xenon guest molecules are only very weakly bound to the framework. This can be observed by the discontinuities in the QHA data points. Second, due to the weak bonding of xenon molecules to the framework, the xenon molecules are able

to sample the entire pore of the framework, giving rise to large entropy contributions. However, this high translational degree of freedom is not captured by QHA in which the xenon atoms are restricted to a small region around their equilibrium positions.

Figure R3.3: Energy profiles for MIL-53(Al) containing 4 xenon atoms computed with the force field from the main manuscript. The red dots (FF at 0K) represent internal energies of optimized structures without thermal corrections, the orange curve represents the internal energy at 300K calculated from the MD simulations. The blue dots (QHA-FF at 300K) and green curve (MD Free energy FF at 300K) shows the free energy computed within the quasi-harmonic approximation and with thermodynamic integration respectively.

We added following sentence in the main manuscript on Page 10 :

"Alternatively free energy profiles may be obtained in the quasi-harmonic approximation. This was done for some materials under study in this paper, more information can be found in section 10 of the SI"

4) In the manuscript, the first time the authors state that the work is based on molecular dynamics simulations is at page 5. Before that, they generically use "microscopic approach". That's true, but in my opinion, the level of theory adopted in the "microscopic approach" should be clearly specified at the beginning of the paper.

We changed the sentence at the bottom of page 3 towards :

"Herein we present a microscopic approach based on classical molecular simulations, to construct the Helmholtz free energy and to uniquely determine the macroscopic response of a material upon stimuli such as mechanical pressure, temperature and adsorbed guest molecules."

- 1 Choi, H. J., Dinca, M. & Long, J. R. Broadly hysteretic H₂ adsorption in the microporous metal-organic framework Co(1,4-benzenedipyrzolate). *J. Am. Chem. Soc.*, 2008, **130**, 7848-7850
- 2 Salles, F., Maurin, G., Serre, C., Llewellyn, P.L., Knöfel, C., Choi, H.J., Filinchuk, Y., Oliviero, L., Vimont, A., Long, J.R. & Férey, G., Multistep N₂ Breathing in the Metal-Organic Framework Co(1,4-benzenedipyrzolate). *J. Am. Chem. Soc.*, 2010, **132**, 13782-13788
- 3 Mason, J. A., Oktawiec, J., Taylor, M.K., Hudson, M.R., Rodriguez, J., Bachman, J.E., Gonzalez, M.I., Cervellino, A., Guagliardi, A., Brown, C.M., Llewellyn, P.L., Masciocchi, N. & Long, J.R. Methane storage in flexible metal-organic frameworks with intrinsic thermal management. *Nature*, 2015, **527**, 357-361
- 4 Vanduyfhuys, L., Ghysels, A., Rogge, S. M. J., Demuyne, R. & Van Speybroeck, V. Semi-analytical mean-field model for predicting breathing in metal-organic frameworks. *Mol. Simulat.*, 2015, **41**, 1311-1328
- 5 Boutin, A., Springuel-Huet, M.-A., Nossov, A., Gédéon, A., Loiseau, T., Volkringer, C., Férey, G., Coudert, F.-X. & Fuchs, A.H., Breathing Transitions in MIL-53(Al) Metal-Organic Framework Upon Xenon Adsorption. *Angew. Chem.-Int. Edit.*, 2009, **48**, 8314-8317
- 6 Vanduyfhuys, L., Ghysels, A., Rogge, S. M. J., Demuyne, R. & Van Speybroeck, V. Semi-analytical mean-field model for predicting breathing in metal-organic frameworks. *Mol. Simulat.*, 2015, **41**, 1311-1328
- 7 Stoeck, U., Krause, S., Bon, V., Senkovska, I. & Kaskel, S. A highly porous metal-organic framework, constructed from a cuboctahedral super-molecular building block, with exceptionally high methane uptake. *Chem. Commun.*, 2012, **48**, 10841-10843
- 8 Krause, S., Bon, V., Senkovska, I., Stoeck, U., Wallacher, D., Többsen, D.M., Zander, S., Pillai, R.S., Maurin, G., Coudert, F.-X. & Kaskel, S., A pressure-amplifying framework material with negative gas adsorption transitions. *Nature*, 2016, **532**, 348-352
- 9 Evans, J. D., Bocquet, L. & Coudert, F. X. Origins of Negative Gas Adsorption. *Chem*, 2016, **1**, 873-886
- 10 Rogge, S. M. J., Vanduyfhuys, L., Ghysels, A., Waroquier, M., Verstraelen, T., Maurin, G., Van Speybroeck, V., A Comparison of Barostats for the Mechanical Characterization of Metal-Organic Frameworks. *J. Chem. Theory Comput.*, 2015, **11**, 5583-5597
- 11 Walker, A. M., Civalieri, B., Slater, B., Mellot-Draznieks, C., Corà, F., Zicovich-Wilson, C.M., Román-Pérez, G., Soler, J.M., & Gale, J.D., Flexibility in a Metal-Organic Framework Material Controlled by Weak Dispersion Forces: The Bistability of MIL-53(Al). *Angew. Chem.-Int. Edit.*, 2010, **49**, 7501-7503
- 12 Butler, K. T., Walsh, A., Cheetham, A. K. & Kieslich, G. Organised chaos: entropy in hybrid inorganic-organic systems and other materials. *Chem. Sci.*, 2016, **7**, 6316-6324
- 13 Hoja, J., Reilly, A. M. & Tkatchenko, A. First-principles modeling of molecular crystals: structures and stabilities, temperature and pressure. *Wiley Interdiscip. Rev.-Comput. Mol. Sci.*, 2017, **7**, 26
- 14 Sharp, K. Calculation of Molecular Entropies Using Temperature Integration. *J. Chem. Theory Comput.*, 2013, **9**, 1164-1172
- 15 Joos, L., Lejaeghere, K., Huck, J. M., Van Speybroeck, V. & Smit, B. Carbon capture turned upside down: high-temperature adsorption & low-temperature desorption (HALD). *Energy Environ. Sci.*, 2015, **8**, 2480-2491

- 16 Beurroies, I., Boulhout, M., Llewellyn, P.L., Kuchta, B., Féray, G., Serre, C., Denoyel, R., Using Pressure to Provoke the Structural Transition of Metal-Organic Frameworks. *Angew. Chem.-Int. Edit.*, 2010, **49**, 7526-7529
- 17 Hobday, C. L., Marshall, R.J., Murphie, C.F., Sotelo, J., Richards, T., Allan, D.R., Düren, T., Coudert, F.-X., Forgan, R.S., Morrison, C.A., Moggach, S.A. & Bennett, T.D., A Computational and Experimental Approach Linking Disorder, High-Pressure Behavior, and Mechanical Properties in UiO Frameworks. *Angew. Chem.-Int. Edit.*, 2016, **55**, 2401-2405
- 18 Devautour-Vinot, S., Maurin, G., Henn, F., Serre, C., Devic, T. & Férey, G., Estimation of the breathing energy of flexible MOFs by combining TGA and DSC techniques. *Chem. Commun.*, 2009, **0**, 2733-2735
- 19 Titov, K., Zeng, Z., Ryder, M.R., Chaudhari, A.K., Civalieri, B., Kelley, C.S., Frogley, M.D., Cinque, G., & Tan, J.-C., Probing Dielectric Properties of Metal-Organic Frameworks: MIL-53(Al) as a Model System for Theoretical Predictions and Experimental Measurements via Synchrotron Far- and Mid-Infrared Spectroscopy. *J. Phys. Chem. Lett.*, 2017, **8**, 5035-5040
- 20 R. Batten, S., F. Hoskins, B. & Robson, R. Structures of [Ag(tcm)], [Ag(tcm)(phz)_{1/2}] and [Ag(tcm)(pyz)] (tcm=tricyanomethanide, C(CN)₃-, phz=phenazine, pyz=pyrazine). *New J. Chem.*, 1998, **22**, 173-175
- 21 Ambrosetti, A., Ferri, N., DiStasio, R. A. & Tkatchenko, A. Wavelike charge density fluctuations and van der Waals interactions at the nanoscale. *Science*, 2016, **351**, 1171-1176
- 22 Bjorkman, T., Gulans, A., Krasheninnikov, A. V. & Nieminen, R. M. van der Waals Bonding in Layered Compounds from Advanced Density-Functional First-Principles Calculations. *Phys. Rev. Lett.*, 2012, **108**, 235502-235506
- 23 Bennett, T. D. & Cheetham, A. K. Amorphous Metal-Organic Frameworks. *Accounts Chem. Res.*, 2014, **47**, 1555-1562
- 24 Triguero, C., Coudert, F.-X., Boutin, A., Fuchs, A. H. & Neimark, A. V. Mechanism of Breathing Transitions in Metal-Organic Frameworks. *J. Phys. Chem. Lett.*, 2011, **2**, 2033-2037
- 25 Kmiecik, S., Gront, D., Kolinski, M., Wieteska, L., Dawid, A.E. & Kolinski, A., Coarse-Grained Protein Models and Their Applications. *Chem. Rev.*, 2016, **116**, 7898-7936
- 26 Durholt, J. P., Galvelis, R. & Schmid, R. Coarse graining of force fields for metal-organic frameworks. *Dalton T.*, 2016, **45**, 4370-4379
- 27 Rogge, S. M. J., Wieme, J., Vanduyfhuys, L., Vandenbrande, S., Maurin, G., Verstraelen, T., Waroquier, M. & Van Speybroeck, V., Thermodynamic Insight in the High-Pressure Behavior of UiO-66: Effect of Linker Defects and Linker Expansion. *Chem. Mater.*, 2016, **28**, 5721-5732
- 28 Grosch, J. S. & Paesani, F. Molecular-Level Characterization of the Breathing Behavior of the Jungle-Gym-type DMOF-1 Metal-Organic Framework. *J. Am. Chem. Soc.*, 2012, **134**, 4207-4215
- 29 Wang, J. M., Wolf, R. M., Caldwell, J. W., Kollman, P. A. & Case, D. A. Development and testing of a general amber force field. *J. Comput. Chem.*, 2004, **25**, 1157-1174
- 30 Zhao, Y. & Truhlar, D. G. The M06 suite of density functionals for main group thermochemistry, thermochemical kinetics, noncovalent interactions, excited states, and transition elements: two new functionals and systematic testing of four M06-class functionals and 12 other functionals. *Theor. Chem. Acc.*, 2008, **120**, 215-241
- 31 Breneman, C. M. & Wiberg, K. B. Determining Atom-Centered Monopoles from Molecular Electrostatic Potentials - the Need for High Sampling Density in Formamide Conformational-Analysis. *J. Comput. Chem.*, 1990, **11**, 361-373
- 32 Addicoat, M. A., Vankova, N., Akter, I. F. & Heine, T. Extension of the Universal Force Field to Metal-Organic Frameworks. *J. Chem. Theory Comput.*, 2014, **10**, 880-891
- 33 Boyd, P. G., Moosavi, S. M., Witman, M. & Smit, B. Force-Field Prediction of Materials Properties in Metal-Organic Frameworks. *J. Phys. Chem. Lett.*, 2017, **8**, 357-363
- 34 Tafipolsky, M., Amirjalayer, S. & Schmid, R. Ab initio parametrized MM3 force field for the metal-organic framework MOF-5. *J. Comput. Chem.*, 2007, **28**, 1169-1176
- 35 Tafipolsky, M. & Schmid, R. Systematic First Principles Parameterization of Force Fields for Metal-Organic Frameworks using a Genetic Algorithm Approach. *J. Phys. Chem. B*, 2009, **113**, 1341-1352
- 36 Vanduyfhuys, L., Vandenbrande, S., Verstraelen, T., Schmid, R., Waroquier, M. & Van Speybroeck, V., QuickFF: A Program for a Quick and Easy Derivation of Force Fields for Metal-Organic Frameworks from Ab Initio Input. *J. Comput. Chem.*, 2015, **36**, 1015-1027

- 37 Allinger, N. L., Yuh, Y. H. & Lii, J. H., Molecular Mechanics - the MM3 Force-Field for Hydrocarbons.1., *J. Am. Chem. Soc.*, 1989, **111**, 8551-8566

Reviewer #1 (Remarks to the Author):

After looking at the revised manuscript and the corrections the authors made to my and the other referees' concerns, I support publication of the manuscript in its current form.

I am particularly pleased that in the revised version the authors incorporated Co(bdp) and DUT-49(Cu) in their discussion. I am convinced that the developed model will be of large interest to computational scientists as well as experimentalists working in this highly dynamic research area. In particular, the developed model will be highly valuable in understanding and assessing the behaviour of (new) soft MOFs and it will be interesting to see how the model will cope with future developments in the field.

Reviewer #2 (Remarks to the Author):

- They tried to apply their approach to 1D or 2D structures but their conclusion is that the current MM3 force field is not suitable to describe such weaker-interacted systems. This is disappointment because many soft MOFs are based on lower-dimensional systems; however, I found many efforts to improve the manuscript regarding to this point in the revised manuscript and I thought this is okay. I suggest that they should add a short description that the current strategy is not fully applicable for 1D or 2D systems with reasons to evoke next challenge.

- I found several responses by the reviewer 2 regarding to the amorphous MOFs and modification of Figure 2. The prediction of softness/flexibility of MOF architecture related to the disorder/amorphous nature is of very challenging I know but surely be the next level challenge. I satisfied these revisions.

- about the point 3 raised by the reviewer 2: I understand how difficult to calculate the system containing both weak-interaction (H-bond, VDW, etc.) and multi-ligand system. I thought finding of meta-stable state which has not been found by experimental approach is crucial for the community with respect to the application (even for high-symmetry MOF systems). I suppose the authors found some "new" meta-stable states for several MOFs in this work and it is better to highlight the new finding by the theoretical approach. Currently the hot topic is moving to the precise control of softness/flexibility in MOFs by tuning the domain size, ligand system, etc. If they expand the methodology to contribute to such complicated systems, this is useful indeed.

I consider the revised version of this manuscript has many additional data/discussion and their responses are reliable to try significant improvement for this article. I found general novelty and broad interest for materials science and recommend for publishing.

Reviewer #3 (Remarks to the Author):

Authors properly revised the manuscript according to my suggestions and comments. Furthermore, authors carefully answered all questions raised by other reviewers. They also did a lot of additional work to fulfill all requirements. I believe that their efforts have improved the overall high quality of the paper. As I wrote in my previous report, I think that the work by V. Spreybroeck and co-workers is of general interest and suitable for publication on Nature Communications. Therefore, I suggest to accept the manuscript as is.

Just a minor point.

In Figure 1, I would suggest to add a label "Framework" to the column that shows the framework of the MOFs. In addition, one could also label the two sub-columns corresponding to the column "Framework" as "expanded" and "contracted".

Reply to Reviewer #1

After looking at the revised manuscript and the corrections the authors made to my and the other referees' concerns, I support publication of the manuscript in its current form.

I am particularly pleased that in the revised version the authors incorporated Co(bdp) and DUT-49(Cu) in their discussion. I am convinced that the developed model will be of large interest to computational scientists as well as experimentalists working in this highly dynamic research area. In particular, the developed model will be highly valuable in understanding and assessing the behaviour of (new) soft MOFs and it will be interesting to see how the model will cope with future developments in the field.

We thank the reviewer for his/her appreciation of our revised paper.

Reply to Reviewer #2

- They tried to apply their approach to 1D or 2D structures but their conclusion is that the current MM3 force field is not suitable to describe such weaker-interacted systems. This is disappointment because many soft MOFs are based on lower-dimensional systems; however, I found many efforts to improve the manuscript regarding to this point in the revised manuscript and I thought this is okay. I suggest that they should add a short description that the current strategy is not fully applicable for 1D or 2D systems with reasons to evoke next challenge.

- I found several responses by the reviewer 2 regarding to the amorphous MOFs and modification of Figure 2. The prediction of softness/flexible of MOF architecture related to the disorder/amorphous nature is of very challenging I know but surely be the next level challenge. I satisfied these revisions.

- about the point 3 raised by the reviewer 2: I understand how difficult to calculate the system containing both weak-interaction (H-bond, VDW, etc.) and multi-ligand system. I thought finding of meta-stable state which has not been found by experimental approach is crucial for the community with respect to the application (even for high-symmetry MOF systems). I suppose the authors found some "new" meta-stable states for several MOFs in this work and it is better to highlight the new finding by the theoretical approach.

Currently the hot topic is moving to the precise control of softness/flexibility in MOFs by tuning the domain size, ligand system, etc. If they expand the methodology to contribute to such complicated systems, this is useful indeed.

I consider the revised version of this manuscript has many additional data/discussion and their responses are reliable to try significant improvement for this article. I found general novelty and broad interest for materials science and recommend for publishing.

We thank the reviewer for his/her appreciation of our revised paper and the useful suggestions. With regard to the first point, we slightly adjusted the last sentence of the 'Force field derivation' subsection in the 'Methods' section to meet the suggestion of the reviewer:

"For some materials such as **one- and** two-dimensional coordination polymers, force field development might be very challenging due to critical dependence on the non-bonding interactions between the layers or the chains **and more dedicated research might be required to apply the methodology on these systems.**"

Reply to Reviewer #3

Authors properly revised the manuscript according to my suggestions and comments. Furthermore, authors carefully answered all questions raised by other reviewers. They also did a lot of additional work to fulfill all requirements. I believe that their efforts have improved the overall high quality of the paper.

As I wrote in my previous report, I think that the work by V. Spreybroeck and co-workers is of general interest and suitable for publication on Nature Communications.

Therefore, I suggest to accept the manuscript as is.

Just a minor point.

In Figure 1, I would suggest to add a label "Framework" to the column that shows the framework of the MOFs. In addition, one could also label the two sub-columns corresponding to the column "Framework" as "expanded" and "contracted".

We thank the reviewer for his/her appreciation of our revised paper. We also thank the reviewer for the suggestions regarding Figure 1, which indeed increase the clarity of the figure. However, we preferred the label 'open' instead of 'expanded' to be consistent with the terminology in the manuscript. The updated figure is shown below.